# FERMI: Fair Empirical Risk Minimization Via Exponential Rényi Mutual Information

## Abstract

Several notions of fairness, such as demographic parity and equal opportunity, are defined based on statistical independence between a predicted target and a sensitive attribute. In machine learning applications, however, the data distribution is unknown to the learner and statistical independence is not verifiable. Hence, the learner could only resort to empirical evaluation of the degree of fairness violation. Many fairness violation notions are defined as a divergence/distance between the joint distribution of the target and sensitive attributes and the Kronecker product of their marginals, such as Rényi correlation, mutual information, $L_\infty$ distance, to name a few. In this paper, we propose another notion of fairness violation, called Exponential Rényi Mutual Information (ERMI) between sensitive attributes and the predicted target. We show that ERMI is a strong fairness violation notion in the sense that it provides an upper bound guarantee on all of the aforementioned notions of fairness violation. We also propose the Fair Empirical Risk Minimization via ERMI regularization framework, called FERMI. Whereas existing in-processing fairness algorithms are deterministic, we provide a stochastic optimization method for solving FERMI that is amenable to large-scale problems. In addition, we provide a batch (deterministic) method to solve FERMI. Both of our proposed algorithms come with theoretical convergence guarantees. Our experiments show that FERMI achieves the most favorable tradeoffs between fairness violation and accuracy on test data across different problem setups, even when fairness violation is measured in notions other than ERMI.

## 1 Introduction

Ensuring that decisions made using machine learning algorithms are fair to different subgroups is of utmost importance. Without any mitigation strategy, machine learning algorithms may result in discrimination against certain subgroups based on sensitive attributes, such as gender or race, even if such discrimination is absent in the training data (Datta et al., 2015; Sweeney, 2013; Bolukbasi et al., 2016; Angwin et al., 2016; du Pin Calmon et al., 2017b; Feldman et al., 2015; Hardt et al., 2016; Fish et al., 2016; Woodworth et al., 2017; Zafar et al., 2017; Bechavod & Ligett, 2017; Kearns et al., 2018). To remedy such discrimination issues, several notions for imposing algorithmic fairness have been proposed in the literature.

A learning machine satisfies the *demographic parity* notion, if the predicted target is independent of the sensitive attributes (Dwork et al., 2012). Promoting demographic parity can lead to poor performance, especially if the true outcome is not independent of the sensitive attributes. To remedy this, (Hardt et al., 2016) proposed *equalized odds* to ensure that the predicted target is conditionally independent of the sensitive attributes given the true label. A further relaxed version of this notion is *equal opportunity* which is satisfied if predicted target is conditionally independent of sensitive attributes given that the true label is in an advantaged class (Hardt et al., 2016). *Note that the inherent assumption in such conditional notions is that the true labels are unbiased. These notions suffer from a potential amplification of the inherent biases that may exist in the targets/labels in the training data (e.g., data collection bias). Tackling such bias is beyond the scope of this work.*

In practice, the learner cannot empirically verify independence of random variables, and hence cannot verify demographic parity, equalized odds, or equal opportunity. This has led the machine learning community to define several notions of fairness violation that quantify the degree of independence between random variables, e.g., demographic parity/equalized odds, $L_\infty$ distance (Dwork et al., 2012; Hardt et al., 2016), mutual information (Kamishima et al., 2011; Rezaei et al., 2020;

Steinberg et al., 2020; Zhang et al., 2018; Cho et al., 2020), Pearson correlation (Zafar et al., 2017), false positive/negative rates (Bechavod & Ligett, 2017), Hilbert Schmidt independence criterion (HSIC) (Pérez-Suay et al., 2017), and Rényi correlation (Baharlouei et al., 2020; Grari et al., 2020; 2019), to name a few. In this paper, we define yet another notion of fairness violation, called exponential Rényi mutual information (ERMI). We show that ERMI is easy to compute empirically and prove that ERMI provides an upper bound on the existing notions of fairness violation such as demographic parity, equalized odds, and equal opportunity.

Given a notion of fairness violation, it is still not straightforward to train an algorithm that satisfies a fairness violation constraint (Cotter et al., 2019). The fairness-promoting machine learning algorithms can be categorized in three main classes: *pre-processing*, *post-processing*, and *in-processing* methods. Pre-processing algorithms (Feldman et al., 2015; Zemel et al., 2013; du Pin Calmon et al., 2017b) transform the biased data features to a new space in which the labels and sensitive attributes are statistically independent. This transform is oblivious to the training procedure. Post-processing approaches (Hardt et al., 2016; Pleiss et al., 2017) mitigate the discrimination of the classifier by altering the the final decision, e.g., by changing the thresholds on soft labels, or reassigning the labels to impose notions of fairness. In-processing approaches focus on the training procedure and impose the notions of fairness as constraints or regularization terms in the optimization procedure. Several regularization-based methods are proposed in the literature to impose measures of fairness to decision-trees (Kamiran et al., 2010; Raff et al., 2018; Aghaei et al., 2019), support vector machines (Donini et al., 2018), neural networks (Grari et al., 2020), or (logistic) regression models (Zafar et al., 2017; Berk et al., 2017; Taskesen et al., 2020; Chzhen & Schreuder, 2020; Baharlouei et al., 2020; Jiang et al., 2020; Grari et al., 2019). To the best of our knowledge, existing in-processing methods are all deterministic, making them impractical for large-scale problems. Furthermore, most in-processing methods (with the exception of (Baharlouei et al., 2020)) are designed for problems in which the sensitive attribute and/or the target is binary. In this paper, we introduce a new fair empirical risk minimization framework via ERMI regularization, and call it FERMI. We provide novel batch and *stochastic* gradient-based methods with guarantees for solving FERMI and demonstrate their effectiveness on multiple numerical experiments, which include a large-scale problem and problem with both non-binary sensitive attributes and targets. We show that FERMI can be used to achieve the most favorable tradeoffs between performance and fairness, even if fairness violation is measured in notions other than ERMI.

## 2 $(Z, \mathcal{Z})$-FAIRNESS: A GENERAL NOTION OF FAIRNESS

We consider a learner who trains a model to predict a target, $\widehat{Y}$, e.g., whether or not to extend a loan, supported on $\mathcal{Y}$ which can be discrete or continuous. The prediction is made using a set of features, $\mathbf{X}$, e.g., financial history features, length of credit, and amount of debt. We also assume that there is a set of discrete sensitive attributes, $S$, e.g., race and sex, supported on $\mathcal{S}$, associated with each sample. Further, let $\mathcal{A} \subseteq \mathcal{Y}$ denote an advantaged outcome class, e.g., the outcome where a loan is extended. Next, we will present the main fairness notion considered in this paper, which generalizes several existing ones.

**Definition 1** ($(Z, \mathcal{Z})$-fairness)**.** *Given a random variable $Z$, let $\mathcal{Z}$ be a subset of values that $Z$ can take. We say that a learning machine satisfies $(Z, \mathcal{Z})$-fairness if for every $z \in \mathcal{Z}$, $\widehat{Y}$ is conditionally independent of $S$ given $Z = z$. More precisely,*

$$p_{\widehat{Y}, S|Z}(\widehat{y}, s|z) = p_{\widehat{Y}|Z}(\widehat{y}|z)p_{S|Z}(s|z) \quad \forall \widehat{y} \in \mathcal{Y}, s \in \mathcal{S}, z \in \mathcal{Z}. \tag{1}$$

Notice that $(Z, \mathcal{Z})$-fairness recovers several important existing notions of fairness as special cases:

1. $(Z, \mathcal{Z})$-fairness recovers demographic parity (Dwork et al., 2012) if $Z = 0$ and $\mathcal{Z} = \{0\}$. In this case, conditioning on $Z$ has no effect, and hence $(0, \{0\})$ fairness is equivalent to the independence between $\widehat{Y}$ and $S$, i.e., demographic parity (see Definition 7, Appendix A).
2. $(Z, \mathcal{Z})$-fairness recovers equalized odds (Hardt et al., 2016) if $Z = Y$ and $\mathcal{Z} = \mathcal{Y}$. In the case, $Z \in \mathcal{Z}$ is trivially satisfied and could be dropped. Hence, conditioning on $Z$ is equivalent to conditioning on $Y$, which recovers the equalized odds notion of fairness, i.e., conditional independence of $\widehat{Y}$ and $S$ given $Y$ (see Definition 8, Appendix A).
3. $(Z, \mathcal{Z})$-fairness recovers equal opportunity (Hardt et al., 2016) if $Z = Y$ and $\mathcal{Z} = \mathcal{A}$. This is also similar to the previous case with $\mathcal{Y}$ replaced with $\mathcal{A}$ (see Definition 9, Appendix A).

Demographic parity amounts to requiring equality of outcomes across sensitive groups. However, it can result in poor performance of a learned model, particularly if the true outcome $Y$ is not independent of $S$. Equalized odds and equal opportunity remedy this issue by relaxing the independece constraint (Hardt et al., 2016). In the binary classification setting with binary sensitive attributes (e.g., male or female), equalized odds ensures equality of false negatives and false positives of a classifier across sensitive groups. Equal opportunity is a further relaxation of the equalized odds criterion, which, in the binary setting, requires just equality of false negatives across sensitive groups. For example, this could be used to enforce that a face recognition software does not falsely classify people of one race as criminals more often than people of other races.

Note that verifying $(Z, \mathcal{Z})$-fairness requires having access to the joint distribution of random variables $(Z, \widehat{Y}, S)$. This joint distribution is unavailable to the learner in the context of machine learning, and hence the learner would resort to empirical estimation of the amount of violation of independence. In the next section, we propose exponential Rényi mutual information as a notion of the violation of $(Z, \mathcal{Z})$-fairness, and show that it is a stronger notion compared to several existing fairness violation notions, including demographic parity $L_\infty$ distance (Kearns et al., 2018), equalized odds $L_\infty$ distance (Hardt et al., 2016), and equal opportunity $L_\infty$ distance (Hardt et al., 2016).

## 3 EXPONENTIAL RÉNYI MUTUAL INFORMATION

In this section, we define ERMI and show that several existing fairness violation notions (which are mostly instances of $f$-divergences or distance metrics between the joint probability distribution and the Kronecker product of the marginals) are also upper bounded by ERMI, implying that ERMI is a stronger notion of fairness violation. In particular, this means that if ERMI is small, then we automatically obtain guarantees that all other notions of fairness violation must be small as well. We present all definitions and results for general $(Z, \mathcal{Z})$ fairness notion, which requires careful extension of several existing notions to the conditional case. These definitions and results will simplify significantly when $Z = 0$ and $\mathcal{Z} = \{0\}$, which will eliminate all conditional expectations.

**Definition 2** (ERMI – exponential Rényi mutual information). *We define the exponential Rényi mutual information between $\widehat{Y}$ and $S$ given $Z \in \mathcal{Z}$ as*

$$D_R(\widehat{Y}; S | Z \in \mathcal{Z}) := \mathbb{E}_{Z, \widehat{Y}, S} \left\{ \frac{p_{\widehat{Y}, S|Z}(\widehat{Y}, S|Z)}{p_{\widehat{Y}|Z}(\widehat{Y}|Z) p_{S|Z}(S|Z)} \middle| Z \in \mathcal{Z} \right\} - 1. \tag{2}$$

In Appendix B, we unravel the definition for the special cases of interest corresponding to the existing notions of fairness. We also discuss that ERMI is the $\chi^2$-divergence (which is an $f$-divergence) between the joint distribution, $p_{\widehat{Y}, S|Z}$, and the Kronecker product of marginals, $p_{\widehat{Y}|Z} \otimes p_{S|Z}$. In particular, ERMI is non-negative, and zero if and only if $(Z, \mathcal{Z})$-fairness is satisfied. Hence, ERMI is a valid notion of fairness violation.

**Definition 3** (Rényi mutual information (Rényi, 1961)). *Let the Rényi mutual information of order $\alpha > 1$ between random variables $\widehat{Y}$ and $S$ given $Z \in \mathcal{Z}$ be defined as:*

$$I_\alpha(\widehat{Y}; S | Z \in \mathcal{Z}) := \frac{1}{\alpha - 1} \log \left( \mathbb{E}_{Z, \widehat{Y}, S} \left\{ \left( \frac{p_{\widehat{Y}, S|Z}(\widehat{Y}, S|Z)}{p_{\widehat{Y}|Z}(\widehat{Y}|Z) p_{S|Z}(S|Z)} \right)^{\alpha - 1} \middle| Z \in \mathcal{Z} \right\} \right), \tag{3}$$

*which generalizes Shannon mutual information*

$$I_1(\widehat{Y}; S | Z \in \mathcal{Z}) := \mathbb{E}_{Z, \widehat{Y}, S} \left\{ \log \left( \frac{p_{\widehat{Y}, S|Z}(\widehat{Y}, S|Z)}{p_{\widehat{Y}|Z}(\widehat{Y}|Z) p_{S|Z}(S|Z)} \right) \middle| Z \in \mathcal{Z} \right\}, \tag{4}$$

*and recovers it as $\lim_{\alpha \to 1^+} I_\alpha(\widehat{Y}; S | Z \in \mathcal{Z}) = I_1(\widehat{Y}; S | Z \in \mathcal{Z})$.*

Note that $I_\alpha(\widehat{Y}; S | Z \in \mathcal{Z}) \geq 0$ with equality if and only if $(Z, \mathcal{Z})$-fairness is satisfied.

**Theorem 1** (ERMI is stronger than Shannon mutual information). *We have*

$$0 \leq I_1(\widehat{Y}; S | Z \in \mathcal{Z}) \leq I_2(\widehat{Y}; S | Z \in \mathcal{Z}) \leq e^{I_2(\widehat{Y}; S | Z \in \mathcal{Z})} - 1 = D_R(\widehat{Y}; S | Z \in \mathcal{Z}). \tag{5}$$

All proofs are relegated to the appendix. Theorem 1 establishes that ERMI is a stronger notion of fairness in the sense that driving it to zero would also bound the Shannon mutual information. It also shows that ERMI is exponentially related to the Rényi mutual information of order 2.

**Definition 4** (Rényi correlation (Hirschfeld, 1935; Gebelein, 1941; Rényi, 1959))**.** *Let $\mathcal{F}$ and $\mathcal{G}$ be the set of measurable functions such that for random variables $\widehat{Y}$ and $S$, $\mathbb{E}_{\widehat{Y}}\{f(\widehat{Y};z)\} = \mathbb{E}_S\{g(S;z)\} = 0$, $\mathbb{E}_{\widehat{Y}}\{f(\widehat{Y};z)^2\} = \mathbb{E}_S\{g(S;z)^2\} = 1$, for all $z \in \mathcal{Z}$. Rényi correlation is:*

$$\rho_R(\widehat{Y}, S | Z \in \mathcal{Z}) := \sup_{f \in \mathcal{F}, g \in \mathcal{G}} \mathbb{E}_{Z,\widehat{Y},S}\left\{ f(\widehat{Y};Z)g(S;Z) \Big| Z \in \mathcal{Z} \right\}. \tag{6}$$

Rényi correlation generalizes Pearson correlation coefficient

$$\rho(\widehat{Y}, S | Z \in \mathcal{Z}) := \mathbb{E}_Z\left\{ \frac{\mathbb{E}_{\widehat{Y},S}\{\widehat{Y}S|Z\}}{\sqrt{\mathbb{E}_{\widehat{Y}}\{\widehat{Y}^2|Z\}\mathbb{E}_S\{S^2|Z\}}} \Bigg| Z \in \mathcal{Z} \right\} \tag{7}$$

to capture nonlinear dependencies between the random variables by finding functions of random variables that maximize the Pearson correlation coefficient between the random variables. In fact, it is true that $\rho_R(\widehat{Y}, S | Z \in \mathcal{Z}) \geq 0$ with equality if and only if $(Z, \mathcal{Z})$-fairness is satisfied. Due to these favorable properties, Rényi correlation has gained popularity as a measure of fairness violation (Baharlouei et al., 2020; Grari et al., 2020).

**Theorem 2** (ERMI is stronger than Rényi correlation.)**.** *We have*

$$0 \leq |\rho(\widehat{Y}, S | Z \in \mathcal{Z})| \leq \rho_R(\widehat{Y}, S | Z \in \mathcal{Z}) \leq D_R(\widehat{Y}; S | Z \in \mathcal{Z}), \tag{8}$$

*and if $|\mathcal{S}| = 2$, $D_R(\widehat{Y}; S | Z \in \mathcal{Z}) = \rho_R(\widehat{Y}, S | Z \in \mathcal{Z})$.*

Next, we turn to another popular notion of fairness violation and establish similar relationships.

**Definition 5** ($L_q$ fairness violation)**.** *We define the $L_q$ fairness violation for $q \geq 1$ by:*

$$L_q(\widehat{Y}, S | Z \in \mathcal{Z}) := \mathbb{E}_Z\left\{ \left( \int_{\widehat{y} \in \mathcal{Y}_0} \sum_{s \in \mathcal{S}_0} \left| p_{\widehat{Y},S|Z}(\widehat{y}, s|Z) - p_{\widehat{Y}|Z}(\widehat{y}|Z)p_{S|Z}(s|Z) \right|^q dy \right)^{\frac{1}{q}} \Bigg| Z \in \mathcal{Z} \right\}. \tag{9}$$

Note that $L_q(\widehat{Y}, S | Z \in \mathcal{Z}) = 0$ if and only if $(Z, \mathcal{Z})$-fairness is satisfied. In particular, $L_\infty$ fairness violation recovers the demographic parity violation (Kearns et al., 2018, Definition 2.1) if we let $\mathcal{Z} = \{0\}$ and $Z = 0$. It also recovers equal opportunity violation (Hardt et al., 2016) if we let $\mathcal{Z} = \mathcal{A}$ and $Z = Y$. $L_q$ fairness violation generalizes this notion by considering the $L_q$ norm of the difference between the joint distribution $p_{\widehat{Y},S}$ and the Kronecker product of the marginal distributions $p_{\widehat{Y}} \otimes p_S$.

**Theorem 3** (ERMI is stronger than $L_\infty$ fairness violation)**.** *Let $\widehat{Y}$ be a discrete or continuous random variable, and $S$ be a discrete random variable supported on a finite set. Then for any $q \geq 1$,[1]*

$$0 \leq L_q(\widehat{Y}, S | Z \in \mathcal{Z}) \leq \sqrt{D_R(\widehat{Y}, S | Z \in \mathcal{Z})}. \tag{10}$$

The above theorem says that if a method controls ERMI value for imposing fairness, $L_\infty$ demographic parity violation (Kearns et al., 2018), $L_\infty$ equal opportunity violation (Hardt et al., 2016), or $L_\infty$ equalized odds violation (Hardt et al., 2016) is also guaranteed to be bounded.

## 4  FAIR RISK MINIMIZATION VIA ERMI

Our goal is to train a model that balances fairness and accuracy objectives. To this end, we introduce *fair risk minimization through exponential Rényi mutual information* framework defined below:[2]

**Definition 6** (FRMI – fair risk minimization through exponential Rényi mutual information)**.** *To balance fairness and accuracy, we consider the learning objective function:*

$$\min_{\boldsymbol{\theta}} \quad \mathbb{E}_{\mathbf{X},Y,S}\left\{ \ell(\mathbf{X}, Y; \boldsymbol{\theta}) \right\} + \lambda D_R(\widehat{Y}_{\boldsymbol{\theta}}(\mathbf{X}); S), \tag{11}$$

---

[1]Note that a similar relationship with TV norm could be established as well.

[2]In this section, we present all results in the context of $Z = 0$ and $\mathcal{Z} = \{0\}$, leaving off all conditional expectations for clarity of presentation. The results could be generalized for general $(Z, \mathcal{Z})$, as we have used the resulting algorithms for empirical experiments.

*where $\ell$ denotes the loss function, such as $L_2$ loss or cross entropy loss; $\lambda > 0$ is a scalar balancing the accuracy versus fairness objectives; $D_R(\widehat{Y}_{\boldsymbol{\theta}}(\mathbf{X}); S)$ is the notion of ERMI given in Eq. (17); and $\widehat{Y}_{\boldsymbol{\theta}}(\mathbf{X})$ is the output of the learned model (e.g., the output of a classification or a regression task, or the cluster number in a clustering task).*

While $\widehat{Y}_{\boldsymbol{\theta}}(\mathbf{X})$ inherently depends on $\mathbf{X}$ and $\boldsymbol{\theta}$, in the rest of this paper, we sometimes leave the dependence of $\widehat{Y}$ on $\mathbf{X}$ and/or $\boldsymbol{\theta}$ implicit for brevity of notation. Notice that we have also left the dependence of the loss on the predicted outcome $\widehat{Y}$ implicit. FRMI is the objective we should solve if demographic parity is the desired fairness notion; if instead we are interested in equalized odds or equal opportunity, then $D_R(\widehat{Y}, S)$ should be replaced by $D_R(\widehat{Y}, S|Z \in \mathcal{Z})$ for an appropriate $(Z, \mathcal{Z})$ pair, per the discussion in Section 3. Since the theory is very similar in both cases, we stick with FRMI as defined in Definition 6. In practice, the true joint distribution of $(\mathbf{X}, S, Y, \widehat{Y})$ is unknown and we only have $N$ samples at our disposal, making it impossible to solve FRMI. Hence, we turn into fair empirical risk minimization via exponential Rényi mutual information (FERMI) approach. While it is natural to estimate $\mathbb{E}_{\mathbf{X},Y,S}\left\{\ell(\mathbf{X}, Y; \boldsymbol{\theta})\right\}$ through the empirical risk, the estimation of $D_R(\widehat{Y}, S)$ in the objective function in Eq. (11) is not as straightforward. In what follows, we propose two approaches for estimating $D_R(\widehat{Y}, S)$. These two approaches result in two different algorithms for balancing fairness and accuracy, where we discuss the benefits and shortcomings of each.

## 4.1 FERMI VIA EMPIRICAL ESTIMATION OF THE PROBABILITY DISTRIBUTIONS

Let $\{\mathbf{x}_i, s_i, y_i, \widehat{y}_i\}_{i \in [N]}$ denote the features, sensitive attributes, targets, and the predictions of the model parameterized by $\boldsymbol{\theta}$ for samples $i \in [N]$. A natural approach to estimate the objective function in Eq. (11) and learning the parameter $\boldsymbol{\theta}$ is through solving the problem

$$\min_{\boldsymbol{\theta}} \left\{ \frac{1}{N} \sum_{i \in [N]} \ell(\mathbf{x}_i, y_i; \boldsymbol{\theta}) + \lambda \widehat{D}_R(\widehat{Y}_{\boldsymbol{\theta}}; S) \right\}, \tag{12}$$

Here $\widehat{D}_R(\widehat{Y}_{\boldsymbol{\theta}}; S) := \sum_{s \in \mathcal{S}} \int_{\widehat{y} \in \mathcal{Y}} \frac{\widehat{p}_{\widehat{Y},S}(\widehat{y},s)^2}{\widehat{p}_{\widehat{Y}}(\widehat{y})\widehat{p}_S(s)} d\widehat{y} - 1$ is an empirical estimate of $D_R(\widehat{Y}_{\boldsymbol{\theta}}, S)$ where $\widehat{p}_S(s)$, $\widehat{p}_{\widehat{Y}}(\widehat{y})$, and $\widehat{p}_{\widehat{Y},S}(\widehat{y}, s)$ are the empirical estimation of the corresponding probability functions. To make the objective function differentiable with respect to $\boldsymbol{\theta}$, we make the following assumption:

**Assumption 1.** *Assume the sensitive attribute is a deterministic function of the features, i.e., $S = f_s(\mathbf{X})$. This trivially holds if the sensitive attribute is available as part of the features. Further, assume the following soft conditional density:*

$$p_{\widehat{Y}_{\boldsymbol{\theta}},S}(\widehat{y}_{\boldsymbol{\theta}}, s) = \mathbb{E}_{\mathbf{X}} \left\{ \mathbb{1}\{f_s(\mathbf{X}) = s\} \frac{e^{-\tau \ell(\mathbf{X}, \widehat{y}; \boldsymbol{\theta})}}{\int_{y \in \mathcal{Y}} e^{-\tau \ell(\mathbf{X}, y; \boldsymbol{\theta})} dy} \right\}, \tag{13}$$

*where $\tau > 0$ controls the softness of the decision and $\tau \to \infty$ would recover the hard decision made by choosing $y$ that minimizes the loss function.*

Assumption 1 is a generalization of the assumption in RFI (Baharlouei et al., 2020), and is natural in problem instances where the decisions made by the learning algorithm are soft decisions. In particular, logistic regression or neural networks with soft-max layer and cross entropy loss satisfy Assumption 1. We can further find $p_{\widehat{Y}|S}$, $p_{\widehat{Y}}$, and $p_S$ from Eq. (13).

Under this assumption, we will have the following empirical estimate of the joint distribution of the predicted target and the sensitive attribute:

$$\widehat{p}_{\widehat{Y}_{\boldsymbol{\theta}},S}(\widehat{y}_{\boldsymbol{\theta}}, s) := \frac{1}{N} \sum_{i \in [N]} \mathbb{1}\{s_i = s\} \frac{e^{-\tau \ell(\mathbf{x}_i, \widehat{y}; \boldsymbol{\theta})}}{\int_{y \in \mathcal{Y}} e^{-\tau \ell(\mathbf{x}_i, y; \boldsymbol{\theta})} dy}, \tag{14}$$

which could be marginalized to define $\widehat{p}_S(s)$ and $\widehat{p}_{\widehat{Y}}(\cdot)$ as well. These empirical probabilities make $\widehat{D}_R(\widehat{Y}_{\boldsymbol{\theta}}; S)$ a differentiable function of $\boldsymbol{\theta}$. Notice that these empirical quantities converge to the

true distributions for finite $\mathcal{S}$ and $\mathcal{Y}$, however, the sample complexity required for their estimation scales linearly with $|\mathcal{S}|$ and $|\mathcal{Y}|$, which implies an exponential scaling with the number of sensitive attributes. Our stochastic algorithm that will be presented in the next section will aim to remedy this potential problem.

To solve Eq. (12), one can apply the gradient descent algorithm and use the dynamics

$$\boldsymbol{\theta}^{t+1} = \boldsymbol{\theta}^t - \eta \nabla_{\boldsymbol{\theta}} \left\{ \frac{1}{N} \sum_{i \in [N]} \ell(\mathbf{x}_i, y_i; \boldsymbol{\theta}^t) + \lambda \widehat{D}_R(\widehat{Y}_{\boldsymbol{\theta}^t}; S) \right\}, \tag{15}$$

where $\eta > 0$ is the learning rate/step-size. Note that when the sensitive attribute is binary, our algorithm is the same as the one in (Baharlouei et al., 2020) since then $\widehat{D}_R(\widehat{Y}_{\boldsymbol{\theta}^t}; S) = \rho_R(\widehat{Y}, S)$, the Rényi correlation, by Theorem 8. However, in the non-binary case, our algorithm is different from (Baharlouei et al., 2020) in general, and we show in Sec. 5 that it achieves a more favorable fairness-accuracy tradeoff curve. Under standard assumptions on the loss function and the learning rate, one can show that the dynamics in Eq. (15) find an $\epsilon$−stationary point, (i.e., a point with the norm of gradient being smaller than $\epsilon$) in $O(\frac{1}{\epsilon^2})$ iterations (Nesterov, 2013).

**Theorem 4.** *(Informal statement) Gradient descent (i.e. Eq. (15)) converges to the set of $\epsilon$-first order stationary points of the FERMI objective (Eq. (12)) in $O(\frac{1}{\epsilon^2})$ iterations (gradient evaluations).*

While this algorithm achieves the optimal rate of first-order methods for general smooth non-convex optimization problems, the empirical ERMI term in the objective in Eq. (12) is a biased estimator of ERMI in Eq. (11). This bias makes the optimization problem in Eq. (12) not suitable for using stochastic methods. For example, in the extreme case, where only one sample is available to the learner for updating the objective at each turn, $\widehat{D}_R(\cdot; \cdot)$ can be severely biased due to the nonlinearities in how it is defined. In the next subsection, we propose another approach for estimation of $D_R(\cdot; \cdot)$ that results in an unbiased estimator, which is amenable to stochastic optimization.

---

**Algorithm 1** Two-Time Scale SGDA for FERMI

1: **Input**: $\boldsymbol{\theta}^0 \in \mathbb{R}^{d_\theta}$, $W^0 \in \mathcal{W} \subset \mathbb{R}^{k \times m}$, step-sizes $(\eta_\theta, \eta_w)$, mini-batch size $M$, fairness parameter $\lambda \geq 0$, iteration number $T$.
2: **for** $t = 0, 1, \ldots, T$ **do**
3:     Draw a batch $B$ of data points $\{(\mathbf{x}_i, y_i)\}_{i \in B}$
4:     Set $\boldsymbol{\theta}^{t+1} \leftarrow \boldsymbol{\theta}^t - \eta_\theta \left[ \frac{1}{|B|} \sum_{i \in B} \nabla_\theta \ell(\mathbf{x}_i, y_i, \boldsymbol{\theta}) - 2\lambda \nabla_\theta \operatorname{vec}(\widehat{\mathbf{y}}_i(\boldsymbol{\theta}) \widehat{\mathbf{y}}_i(\boldsymbol{\theta})^T)^T \operatorname{vec}(W^T W) + 2\lambda \nabla_\theta \operatorname{vec}(\widehat{\mathbf{y}}_i(\boldsymbol{\theta}) \mathbf{s}_i^T)^T \operatorname{vec}(W^T P_s^{-1/2}) \right]$
5:     Set $W^{t+1} \leftarrow \Pi_{\mathcal{W}} \left( W^t + \eta_w \sum_{i \in B} \left[ -2\lambda W \widehat{\mathbf{y}}_i(\boldsymbol{\theta}) \widehat{\mathbf{y}}_i(\boldsymbol{\theta})^T + 2\lambda P_s^{-1/2} \mathbf{s}_i \widehat{\mathbf{y}}_i(\boldsymbol{\theta}) \right] \right)$
6: **end for**
7: Pick $\hat{t}$ uniformly at random from $\{1, \ldots, T\}$
8: **Return:** $\boldsymbol{\theta}^{\hat{t}}$.

---

### 4.2 STOCHASTIC FERMI

In order to solve the population level objective in Eq. (11) using stochastic methods (such as stochastic gradient descent), one needs to obtain an unbiased estimate of the objective in equation 11, i.e., $\mathbb{E}_{\mathbf{X}, Y, S} \{\ell(\mathbf{X}, Y; \boldsymbol{\theta})\} + \lambda D_R(\widehat{Y}_{\boldsymbol{\theta}}(\mathbf{X}); S)$. Clearly, the empirical average $\frac{1}{|B|} \sum_{i \in B} \ell(\mathbf{x}_i, y_i; \boldsymbol{\theta})$ is an unbiased estimator of $\mathbb{E}_{\mathbf{X}, Y, S} \{\ell(\mathbf{X}, Y; \boldsymbol{\theta})\}$, where $B \subseteq [N]$ is a batch of data points. Thus, to develop a stochastic algorithm, we need to have an unbiased estimator of $D_R(\widehat{Y}_{\boldsymbol{\theta}}(\mathbf{X}); S)$ given a batch of data points $B$. The following Theorem will help us obtain such an estimator.

**Theorem 5.** *For discrete random variables $\widehat{Y}$ and $S$ where $\widehat{Y} \in [m], S \in [k]$, we have*

$$D_R(\widehat{Y}; S) = \max_{W \in \mathbb{R}^{k \times m}} \left\{ -\operatorname{Tr}(W P_{\widehat{y}} W^T) + 2 \operatorname{Tr}(W P_{\widehat{y}, s} P_s^{-1/2}) - 1 \right\}, \tag{16}$$

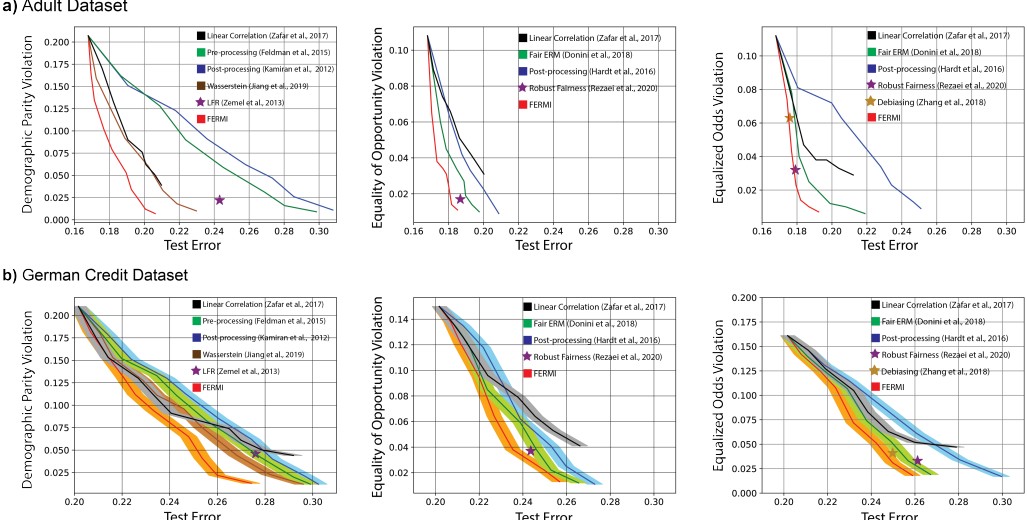

Figure 1: Tradeoff of fairness violation vs test error for different baselines on German Credit and Adult datasets. The desired operation point is the lower left corner where both fairness violation and test error are small. FERMI achieves the best fairness vs performance tradeoff across all baselines.

$$where \; P_{\widehat{y}} = \begin{pmatrix} p_{\widehat{Y}}(1) & & 0 \\ & \ddots & \\ 0 & & p_{\widehat{Y}}(m) \end{pmatrix},$$

$$P_{\widehat{y},s} = \begin{pmatrix} p_{\widehat{Y},S}(1,1) & \dots & p_{\widehat{Y},S}(1,k) \\ \vdots & \ddots & \vdots \\ p_{\widehat{Y},S}(m,1) & \dots & p_{\widehat{Y},S}(m,k) \end{pmatrix}, P_s = \begin{pmatrix} p_S(1) & & 0 \\ & \ddots & \\ 0 & & p_S(k) \end{pmatrix}.$$

Let $\widehat{\mathbf{Y}} \in \{0,1\}^m$ and $\widehat{\mathbf{S}} \in \{0,1\}^k$ be the one hot encoded version of $\widehat{Y}$ and $\widehat{S}$, respectively. Then, the above theorem implies that Eq. (11) can be re-written as

$$\min_{\boldsymbol{\theta}} \max_{W \in \mathbb{R}^{k \times m}} \mathbb{E} \left\{ \ell(\mathbf{X}, Y; \boldsymbol{\theta}) - \mathrm{Tr}(W\widehat{\mathbf{Y}}\widehat{\mathbf{Y}}^T W^T) + 2\mathrm{Tr}(W\widehat{\mathbf{Y}}\mathbf{S}^T P_s^{-1/2}) - 1 \right\}.$$

Hence, given a batch of data points $B$, we can obtain an unbiased estimator of the above objective function by the empirical average $\frac{1}{|B|}\sum_{i \in B} \left\{ \ell(\mathbf{x}_i, y_i; \boldsymbol{\theta}) - \mathrm{Tr}(W\widehat{\mathbf{y}}_i\widehat{\mathbf{y}}_i^T W^T) + 2\mathrm{Tr}(W\widehat{\mathbf{y}}_i\mathbf{s}_i^T P_s^{-1/2}) - 1 \right\}$. This observation leads to the stochastic algorithm presented in Algorithm 1. Notice that this algorithm is based on the assumption that $P_s$ is known. This assumption is practical since the distribution of sensitive attributes (such as male vs female) is known in many applications (or it can be estimated accurately using the training data). The convergence rate of Algorithm 1 is analyzed in Theorem 6.

**Theorem 6.** *(Informal statement) Algorithm 1 converges to the set of $\epsilon$-first order stationary points of the FERMI objective (c.f. Eq. 12) in $O(\frac{1}{\epsilon^4})$ iterations (stochastic gradient evaluations).*

The formal statement of this theorem can be found in Theorem 11 in Appendix D. Notice that while this algorithm has a slower rate of convergence than the batch algorithm, it is stochastic (each iteration is computationally cheap) and amenable to large-scale problems. Note also that a faster convergence rate of $O(\frac{1}{\epsilon^3})$ could be obtained by using the (more complicated) SREDA method of (Luo et al., 2020) instead of SGDA to solve FERMI objective. We omit the details here. In the next section, we numerically evaluate the performance of the algorithms described in this section.

## 5 NUMERICAL EXPERIMENTS

### 5.1 BINARY FAIR CLASSIFICATION WITH A BINARY SENSITIVE ATTRIBUTE

We start the experimental setup for binary classification problem with a binary sensitive attribute. This is a common setup among most existing baseline methods. Per Theorem 2, in this binary

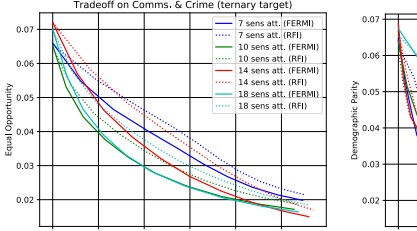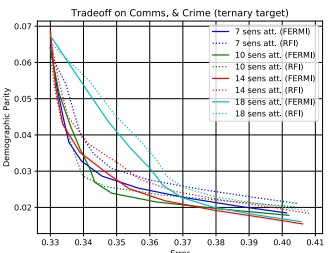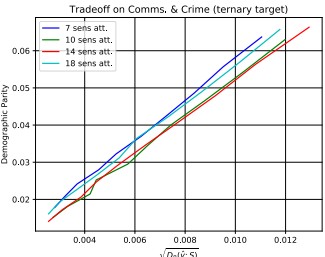

Figure 2: Comparison between FERMI and RFI (Baharlouei et al., 2020). FERMI achieves a better fairness vs performance tradeoff. Moreover, due to computationally expensive operations like performing singular value decomposition (SVD), RFI has poor scalability with the cardinality of sensitive features and target classes.

classification case ERMI is equivalent to Rényi correlation (Baharlouei et al., 2020), and per the discussion in Sec. 4.1, our batch algorithm is exactly the same as RFI (Baharlouei et al., 2020). While we solve FERMI to impose an ERMI regularizer, we still measure fairness violation via popular fairness violation notions, such as conditional demographic parity $L_\infty$ violation (Definition 10), conditional equal opportunity $L_\infty$ violation (Definition 11), and conditional equalized odds violation. In Fig. 1, we report the fairness violation vs error for German Credit and Adult datasets. As can be seen, for all three popular notions of fairness, FERMI achieves the best trade-off between fairness and error probability on the test data. This could be partly due to smoothness of FERMI optimization problem, and partly due to the fact that ERMI upper bounds other fairness notions as discussed in Sec. 3. We will have a more detailed discussion on this in Section 6.

## 5.2 NON-BINARY FAIR CLASSIFICATION WITH A NON-BINARY SENSITIVE ATTRIBUTE

Next, we consider a general classification problem with $|\mathcal{S}| > 2$. In this case, we consider the Communities and Crime dataset, which has 18 binary sensitive attributes in total, and we pick $\{7, 10, 14, 18\}$ sensitive attributes out of those for different experiments, which corresponds to $|\mathcal{S}| \in \{2^7, 2^{10}, 2^{14}, 2^{18}\}$. We discretize the target into three classes $\{\text{high}, \text{medium}, \text{low}\}$ (ternary classification). The only baseline that we are aware of that can handle non-binary classification with non-binary is sensitive attributes is RFI (Baharlouei et al., 2020). The results are presented in Fig. 2, where we use conditional demographic parity $L_\infty$ violation (Definition 10) and conditional equal opportunity $L_\infty$ violation (Definition 11) as the fairness violation notion. As can be seen, FERMI achieves a better tradeoff curve as compared with RFI. It is noteworthy that the per-iteration complexity of FERMI is far less than that of RFI, which requires solving a singular value decomposition at each iteration. Finally, the convergence rate for FERMI given in Theorem 4 guarantees an $O(\frac{1}{\epsilon^2})$ convergence vs $O(\frac{1}{\epsilon^4})$ for RFI (Baharlouei et al., 2020, Theorem 4.1). Empirically, we observed that FERMI converges $\sim$10x faster on this problem instance. Finally, we also show that conditional demographic parity $L_\infty$ violation (Definition 10) and square root of ERMI are approximately linearly related, which further justifies the use of ERMI regularizer in the FERMI framework.

## 5.3 LARGE-SCALE CLASSIFICATION WITH FERMI

In this experiment, we consider the color MNIST dataset (Li & Vasconcelos, 2019) where MNIST digits are colored with different colors drawn from a Gaussian distribution with variance $\sigma$ around a certain average color. It is shown in (Li & Vasconcelos, 2019) that as $\sigma \to 0$, a convolutional network model overfits significantly to the color on this dataset, and hence will not be able to generalize on a regular black and white test set. Our goal in this experiment is to show that FERMI can promote independence between the predicted target and color

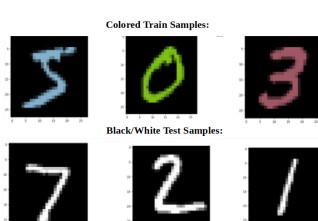

Figure 3: Color MNIST.

(which we use as the sensitive attribute within FERMI) to improve generalization in this setup. This also allows us to examine the scaling of stochastic FERMI when used in convolutional neural networks. We consider $\sigma = 0$, where the test performance is the lowest due to overfitting.

The result of the experiment is presented in Fig. 4. As expected, FERMI results in learning representations that have less dependence on the color, hence leading to better generalization. It is

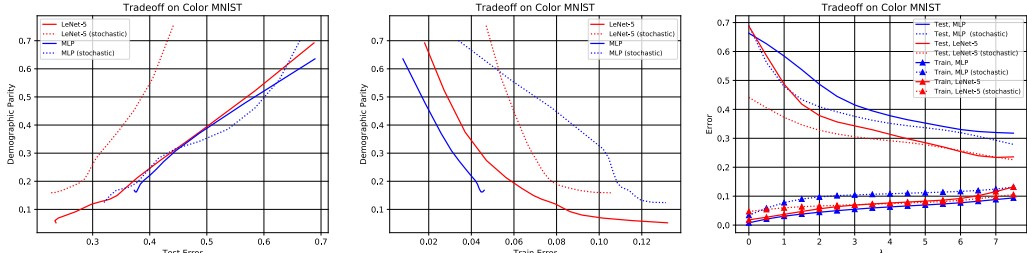

Figure 4: The application of FERMI to Color MNIST dataset (Li & Vasconcelos, 2019). As expected, FERMI achieves a tradeoff between demographic parity $L_\infty$ violation and train error on the colored training samples. On test data, however, reducing the demographic parity violation translates to less dependence on the color, which avoids overfitting and decreases the test error. Finally, as $\lambda$ increases we see that the gap between train error and test error significantly decreases which shows that FERMI can avoid overfitting. For stochastic FERMI, we use a mini-batch of size 512 and achieve a speedup of 100x per iteration. Along with scalability, we also observe that the stochastic variant (Algorithm 1) has better generalization performance (lower test error).

noteworthy that the test error achieved by FERMI when $\sigma = 0$ is 22.6%, as compared to 23.3% obtained using REPAIR (Li & Vasconcelos, 2019) for $\sigma = 0.1$. Further decreasing $\sigma \leq 0.05$ the test error using REPAIR sharply goes above 50%. We were unable to run REPAIR for $\sigma = 0$.

## 6 DISCUSSION & CONCLUDING REMARKS

In this paper, we proposed a new notion of fairness, called exponential Rényi mutual information (ERMI). We showed that ERMI is a strong notion of fairness violation providing guarantees on several other popular notions, namely Pearson correlation, Rényi correlation, Shannon mutual information, Rényi mutual information, and $L_q$ distance violation. We proposed a Fair Empirical Risk Minimization framework with an ERMI regularizer to balance performance and fairness, and called it FERMI. Additionally, we showed that FERMI could be efficiently solved for non-binary sensitive attributes and non-binary target variables. We proposed batch and *stochastic* algorithms for solving FERMI with convergence guarantees for smooth losses. In particular, Algorithm 1 is unique among existing fair algorithms as it is stochastic, making it much more practical for large-scale problems (as demonstrated in Sec. 5.3). This is made possible by Theorem 5, which leads to an unbiased estimator of the gradient of ERMI. It is not at all clear if replacing ERMI by another regularizer, such as Rényi correlation or Shannon mutual information, in Eq. (12) would be amenable to stochastic optimization.

From an experimental perspective, we showed that FERMI leads to better fairness-accuracy tradeoffs than the existing baselines. There are several possible explanations for the superior empirical performance of FERMI compared to existing methods. One possible reason is that the objective function Eq. (12) is easier to optimize than the objectives of competing in-processing methods: ERMI is smooth; and in the discrete case, is equal to the trace of a matrix (see Theorem 8), which is easy to compute. Contrast this with the larger computational overhead of Rényi correlation, for example, which requires finding the second singular value of a matrix. Furthermore, the sample complexity of estimating Rényi mutual information of order 2 (and consequently that of ERMI) scales as $\Theta(\sqrt{|\mathcal{S}|})$ as compared to Shannon mutual information which scales as $\Theta(|\mathcal{S}| / \log |\mathcal{S}|)$ (Acharya et al., 2014). Another possible explanation is that ERMI is a stronger notion of fairness than all of the most widely used fairness notions, as shown in Sec. 3, which might lead to better generalization. Together, these facts suggest that ERMI serves as an efficient and easily optimizable proxy for these other notions, leading to better practical performance regardless of which fairness measure is used. We leave it as future work to rigorously understand which of these (or other) factors are responsible for the favorable performance tradeoffs observed from FERMI. Finally, on the Color MNIST experiment with neural network function approximation, we observed that stochastic FERMI outperforms batch FERMI. In this case, we suspect that the randomness in stochastic FERMI supposedly contributes to its convergence to a local minimum with superior generalization performance compared to batch FERMI (see (Kleinberg et al., 2018) and the references therein).

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

## A    EXISTING NOTIONS OF FAIRNESS

Let $(Y, \widehat{Y}, \mathcal{A}, S)$ denote the true target, predicted target, the advantaged outcome class, and the sensitive attribute, respectively. We review three major notions of fairness.

**Definition 7** (demographic parity (Dwork et al., 2012)). *We say that a learning machine satisfies demographic parity if $\widehat{Y}$ is independent of $S$.*

**Definition 8** (equalized odds (Hardt et al., 2016)). *We say that a learning machine satisfies equalized odds, if $\widehat{Y}$ is conditionally independent of $S$ given $Y$.*

**Definition 9** (equal opportunity (Hardt et al., 2016)). *We say that a learning machine satisfies equal opportunity with respect to $\mathcal{A}$, if $\widehat{Y}$ is conditionally independent of $S$ given $Y = y$ for all $y \in \mathcal{A}$.*

Notice that the equal opportunity as defined here generalizes the definition in (Hardt et al., 2016). It recovers equalized odds if $\mathcal{A} = \mathcal{Y}$, and it recovers equal opportunity of (Hardt et al., 2016) for $\mathcal{A} = \{1\}$ in binary classification.

## B    PROPERTIES AND SPECIAL CASES OF ERMI

Notice that ERMI is in fact the $\chi^2$-divergence between the conditional joint distribution, $p_{\widehat{Y},S}$, and the Kronecker product of conditional marginals, $p_{\widehat{Y}} \otimes p_S$, where the conditioning is on $Z \in \mathcal{Z}$. Further, $\chi^2$-divergence is an $f$-divergence with $f(t) = (t-1)^2$. See (Csiszár & Shields, 2004, Section 4) for a discussion. As an immediate result of this observation and well-known properties of $f$-divergences, we can state the following property of ERMI:

**Remark 7.** $D_R(\widehat{Y}; S | Z \in \mathcal{Z}) \geq 0$ *with equality if and only if for all $z \in \mathcal{Z}$, $\widehat{Y}$ and $S$ are conditionally independent given $Z = z$.*

To further clarify the definition of ERMI, especially as it relates to demographic parity, equalized odds, and equal opportunity, we will unravel the definition explicitly in a few special cases.

First, let $Z = 0$ and $\mathcal{Z} = \{0\}$. In this case, $Z \in \mathcal{Z}$ trivially holds, and conditioning on $Z$ has no effect, resulting in:

$$
\begin{aligned}
D_R(\widehat{Y}; S) := D_R(\widehat{Y}; S | Z \in \mathcal{Z})\Big|_{Z=0, \mathcal{Z}=\{0\}} \\
= \mathbb{E}_{\widehat{Y},S} \left\{ \frac{p_{\widehat{Y},S}(\widehat{Y}, S)}{p_{\widehat{Y}}(\widehat{Y}) p_S(S)} \right\} - 1 \\
= \sum_{s \in S} \int_{\widehat{y} \in \mathcal{Y}} \frac{p_{\widehat{Y},S}(\widehat{y}, s) - p_{\widehat{Y}}(\widehat{y}) p_S(s)}{p_{\widehat{Y}}(\widehat{y}) p_S(s)} p_{\widehat{Y},S}(\widehat{y}, s) d\widehat{y}.
\end{aligned}
\tag{17}
$$

$D_R(\widehat{Y}; S)$ is the notion of ERMI that should be used when the desired notion of fairness is demographic parity. In particular, $D_R(\widehat{Y}; S) = 0$ implies that $\chi^2$ divergence between $p_{\widehat{Y},S}$, and the Kronecker product of marginals, $p_{\widehat{Y}} \otimes p_S$ is zero. This in turn implies that $\widehat{Y}$ and $S$ are independent, which is the definition of demographic parity. We note that when $\widehat{Y}$ and $S$ are discrete, this special case ($Z = 0$ and $\mathcal{Z} = \{0\}$) of ERMI is referred to as $\chi^2$-information in (du Pin Calmon et al., 2017a).

Next, we consider $Z = Y$ and $\mathcal{Z} = \mathcal{Y}$. In this case, $Z \in \mathcal{Z}$ is trivially satisfied, and hence,

$$
\begin{aligned}
D_R(\widehat{Y}; S|Y) := \ & D_R(\widehat{Y}; S|Z \in \mathcal{Z})\Big|_{Z=Y, \mathcal{Z}=\mathcal{Y}} \\
= \ & \mathbb{E}_{Y, \widehat{Y}, S} \left\{ \frac{p_{\widehat{Y}, S|Y}(\widehat{Y}, S|Y)}{p_{\widehat{Y}|Y}(\widehat{Y}|Y) p_{S|Y}(S|Y)} \right\} - 1 \\
= \ & \sum_{s \in \mathcal{S}} \int_{y \in \mathcal{Y}} \int_{\widehat{y} \in \mathcal{Y}} \frac{p_{\widehat{Y}, S|Y}(\widehat{y}, s|y) - p_{\widehat{Y}|Y}(\widehat{y}|y) p_{S|Y}(s|y)}{p_{\widehat{Y}|Y}(\widehat{y}|y) p_{S|Y}(s|y)} p_{Y, \widehat{Y}, S}(y, \widehat{y}, s) d\widehat{y} dy \\
= \ & \sum_{s \in \mathcal{S}} \int_{y \in \mathcal{Y}} \int_{\widehat{y} \in \mathcal{Y}} \frac{p_{\widehat{Y}, S|Y}(\widehat{y}, s|y)^2}{p_{\widehat{Y}|Y}(\widehat{y}|y) p_{S|Y}(s|y)} p_Y(y) d\widehat{y} dy - 1. \tag{18}
\end{aligned}
$$

$D_R(\widehat{Y}; S|Y)$ should be used when the desired notion of fairness is equalized odds. In particular, $D_R(\widehat{Y}; S|Y) = 0$ directly implies the conditional independence of $\widehat{Y}$ and $S$ given $Y$.

Finally, we consider $Z = Y$ and $\mathcal{Z} = \mathcal{A}$. In this case, we have

$$
\begin{aligned}
D_R^{\mathcal{A}}(\widehat{Y}; S|Y) := \ & D_R(\widehat{Y}; S|Z \in \mathcal{Z})\Big|_{Z=Y, \mathcal{Z}=\mathcal{A}} \\
= \ & \mathbb{E}_{Y, \widehat{Y}, S} \left\{ \frac{p_{\widehat{Y}, S|Y}(\widehat{Y}, S|Y)}{p_{\widehat{Y}|Y}(\widehat{Y}|Y) p_{S|Y}(S|Y)} \bigg| Y \in \mathcal{A} \right\} - 1 \\
= \ & \sum_{s \in \mathcal{S}} \int_{y \in \mathcal{A}} \int_{\widehat{y} \in \mathcal{Y}} \frac{p_{\widehat{Y}, S|Y}(\widehat{y}, s|y) - p_{\widehat{Y}|Y}(\widehat{y}|y) p_{S|Y}(s|y)}{p_{\widehat{Y}|Y}(\widehat{y}|y) p_{S|Y}(s|y)} p_Y^{\mathcal{A}}(y) d\widehat{y} dy \\
= \ & \sum_{s \in \mathcal{S}} \int_{y \in \mathcal{A}} \int_{\widehat{y} \in \mathcal{Y}} \frac{p_{\widehat{Y}, S|Y}(\widehat{y}, s|y)^2}{p_{\widehat{Y}|Y}(\widehat{y}|y) p_{S|Y}(s|y)} p_{\widehat{Y}, S|Y}(\widehat{y}, s|y) p_Y^{\mathcal{A}}(y) d\widehat{y} dy - 1, \tag{19}
\end{aligned}
$$

where

$$
p_Y^{\mathcal{A}}(y) := \frac{p_Y(y)}{\int_{y' \in \mathcal{A}} p_Y(y') dy'}. \tag{20}
$$

This notion is what should be used when the desired notion of fairness is equal opportunity. This can be further simplified when the advantaged class is a singleton (which is the case in binary classification). If $Z = Y$ and $\mathcal{Z} = \{y\}$, then

$$
\begin{aligned}
D_R(\widehat{Y}; S|Y = y) := \ & D_R^{\{y\}}(\widehat{Y}; S|Y) \\
= \ & \sum_{s \in \mathcal{S}} \int_{\widehat{y} \in \mathcal{Y}} \frac{p_{\widehat{Y}, S|Y}(\widehat{y}, s|y) - p_{\widehat{Y}|Y}(\widehat{y}|y) p_{S|Y}(s|y)}{p_{\widehat{Y}|Y}(\widehat{y}|y) p_{S|Y}(s|y)} p_{\widehat{Y}, S|Y}(\widehat{y}, s|y) d\widehat{y} \\
= \ & \sum_{s \in \mathcal{S}} \int_{\widehat{y} \in \mathcal{Y}} \frac{p_{\widehat{Y}, S|Y}(\widehat{y}, s|y)^2}{p_{\widehat{Y}|Y}(\widehat{y}|y) p_{S|Y}(s|y)} d\widehat{y} - 1. \tag{21}
\end{aligned}
$$

Finally, we note that we use the notation $D_R(\widehat{Y}; S|Y)$ and $D_R(\widehat{Y}; S|Y = y)$ to be consistent with the definition of conditional mutual information in (Cover & Thomas, 1991).

## C  RELATIONS BETWEEN ERMI AND OTHER FAIRNESS VIOLATION NOTIONS

*Proof of Theorem 1.* We proceed to prove all the (in)equalities one by one:

- $0 \leq I_S(\widehat{Y}; S|Z \in \mathcal{Z})$. This is well known and the proof can be found in any information theory textbook (Cover & Thomas, 1991).

- $I_1(\widehat{Y}; S|Z \in \mathcal{Z}) \leq I_2(\widehat{Y}; S|Z \in \mathcal{Z})$. This is a known property of Rényi mutual information, but we provide a proof for completeness in Lemma 1.

- $I_2(\widehat{Y}; S|Z \in \mathcal{Z}) \leq e^{I_2(\widehat{Y};S|Z \in \mathcal{Z})} - 1$. This follows from the fact that $x \leq e^x - 1$.

- $e^{I_2(\widehat{Y};S)|Z \in \mathcal{Z}} - 1 = D_R(\widehat{Y}; S|Z \in \mathcal{Z})$. This follows from simple algebraic manipulation.

$\square$

**Lemma 1.** *Let $\widehat{Y}, S, Z$ be discrete or continuous random variables. Then:*

*(a) For any $\alpha, \beta \in [1, \infty]$, $I_\beta(\widehat{Y}; S|Z \in \mathcal{Z}) \geq I_\alpha(\widehat{Y}; S|Z \in \mathcal{Z})$ if $\beta > \alpha$.*

*(b) $\lim_{\alpha \to 1^+} I_\alpha(\widehat{Y}; S|Z \in \mathcal{Z}) = I_1(\widehat{Y}; S) := \mathbb{E}_Z \left\{ D_{KL}(p_{\widehat{Y},S|Z} || p_{\widehat{Y}|Z} \otimes p_{S|Z}) \Big| Z \in \mathcal{Z} \right\}$, where $I_1(\cdot; \cdot)$ denotes the Shannon mutual information and $D_{KL}$ is Kullback–Leibler divergence (relative entropy).*

*(c) For all $\alpha \in [1, \infty]$, $I_\alpha(\widehat{Y}; S|Z \in \mathcal{Z}) \geq 0$ with equality if and only if for all $z \in \mathcal{Z}$, $\widehat{Y}$ and $S$ are conditionally independent given $z$.*

*Proof.* *(a)* First assume $0 < \alpha < \beta < \infty$ and that $\alpha, \beta \neq 1$. Define $a = \alpha - 1$, and $b = \beta - 1$. Then the function $\phi(t) = t^{b/a}$ is convex for all $t \geq 0$, so by Jensen's inequality we have:

$$\frac{1}{b} \log \left( \mathbb{E} \left\{ \left( \frac{p(\widehat{Y}, S|Z)}{p(\widehat{Y}|Z)p(S|Z)} \right)^b \Bigg| Z \in \mathcal{Z} \right\} \right) \geq \frac{1}{b} \log \left( \mathbb{E} \left\{ \left( \frac{p(\widehat{Y}, S|Z)}{p(\widehat{Y}|Z)p(S|Z)} \right)^a \Bigg| Z \in \mathcal{Z} \right\}^{b/a} \right)$$
$$= \frac{1}{a} \log \left( \mathbb{E} \left\{ \left( \frac{p(\widehat{Y}, S|Z)}{p(\widehat{Y}|Z)p(S|Z)} \right)^a \Bigg| Z \in \mathcal{Z} \right\} \right). \tag{22}$$

Now suppose $\alpha = 1$. Then by the monotonicity for $\alpha \neq 1$ proved above, we have $I_1(\widehat{Y}; S) = \lim_{\alpha \to 1^-} I_\alpha(\widehat{Y}; S) = \sup_{\alpha \in (0,1)} I_\alpha(\widehat{Y}; S) \leq \inf_{\alpha > 1} I_\alpha(\widehat{Y}; S)$. Also, $I_\infty(\widehat{Y}; S) = \lim_{\alpha \to \infty} I_\alpha(\widehat{Y}; S) = \sup_{\alpha > 0} I_\alpha(\widehat{Y}; S)$.

*(b)* This is a standard property of the cumulant generating function (see (Dembo & Zeitouni, 2009)).

*(c)* It is straightforward to observe that independence implies that Rényi mutual information vanishes. On the other hand, if Rényi mutual information vanishes, then part (a) implies that Shannon mutual information also vanishes, which implies the desired conditional independence. $\square$

*Proof of Theorem 2.* The proof is completed using the following pieces.

- $0 \leq |\rho(\widehat{Y}, S|Z \in \mathcal{Z})| \leq \rho_R(\widehat{Y}, S|Z \in \mathcal{Z})$. This is obvious from the definition of $\rho_R(\widehat{Y}, S|Z \in \mathcal{Z})$.

- $\rho_R(\widehat{Y}, S|Z \in \mathcal{Z}) \leq D_R(\widehat{Y}; S|Z \in \mathcal{Z})$. This follows from Theorem 8.

- Notice that if $|\mathcal{S}| = 2$, Theorem 8 implies that $D_R(\widehat{Y}; S|Z \in \mathcal{Z}) = \rho_R(\widehat{Y}, S|Z \in \mathcal{Z})$.

$\square$

**Theorem 8.** *Suppose that $\mathcal{S} = [k]$. Let the $k \times k$ matrix $P$ be defined as $P = \{P_{ij}\}_{i,j \in [k] \times [k]}$, where*

$$P_{ij} := \frac{1}{\sqrt{p_S(i)p_S(j)}} \int_{y \in \mathcal{Y}} \left( \frac{p_{\widehat{Y},S}(y, i)p_{\widehat{Y},S}(y, j)}{p_{\widehat{Y}}(y)} \right) dy. \tag{23}$$

*Let $1 = \sigma_1 \geq \sigma_2 \geq \ldots \geq \sigma_k \geq 0$ be the eigenvalues of $P$. Then,*

$$\rho_R(\widehat{Y}, S) = \sigma_2, \tag{24}$$

$$D_R(\widehat{Y}; S) = \text{Tr}(P) - 1 = \sum_{i=2}^{k} \sigma_i. \tag{25}$$

*Proof.* Eq. (24) is proved in (Witsenhausen, 1975, Section 3). To prove Eq. (25), notice that

$$\mathrm{Tr}(P) = \sum_{i \in [k]} P_{ii}$$

$$= \sum_{i \in [k]} \frac{1}{p_S(i)} \int_{y \in \mathcal{Y}} \left( \frac{p_{\widehat{Y},S}(y,i)^2}{p_{\widehat{Y}}(y)} \right) dy$$

$$= E_{\widehat{Y},S} \left\{ \left( \frac{p_{\widehat{Y},S}(\widehat{Y},S)}{p_{\widehat{Y}}(\widehat{Y}) p_S(S)} \right) \right\}$$

$$= 1 + D_R(\widehat{Y}; S),$$

which completes the proof. □

*Proof of Theorem 3.* It suffices to prove the inequality for $L_1$, as $L_q$ is bounded above by $L_1$ for all $q \geq 1$. The proof for the case where $Z = 0$ and $\mathcal{Z} = \{0\}$ follows from the following set of inequalities:

$$L_1(\widehat{Y}, S | Z \in \mathcal{Z}) = \sum_{s \in \mathcal{S}} \int_{y \in \mathcal{Y}} \left| p_{\widehat{Y},S}(y,s) - p_{\widehat{Y}}(y) p_S(s) \right| dy \tag{26}$$

$$= \sum_{s \in \mathcal{S}} \int_{y \in \mathcal{Y}} \sqrt{p_{\widehat{Y}}(y) p_S(s)} \frac{\left| p_{\widehat{Y},S}(y,s) - p_{\widehat{Y}}(y) p_S(s) \right|}{\sqrt{p_{\widehat{Y}}(y) p_S(s)}} dy \tag{27}$$

$$\leq \sqrt{\left( \sum_{s \in \mathcal{S}} \int_{y \in \mathcal{Y}} p_{\widehat{Y}}(y) p_S(s) dy \right) \left( \sum_{s \in \mathcal{S}} \int_{y \in \mathcal{Y}} \left( \frac{(p_{\widehat{Y},S}(y,s) - p_{\widehat{Y}}(y) p_S(s))^2}{p_{\widehat{Y}}(y) p_S(s)} \right) \right)} \tag{28}$$

$$\leq \sqrt{\sum_{s \in \mathcal{S}} \int_{y \in \mathcal{Y}} \left( \frac{(p_{\widehat{Y},S}(y,s) - p_{\widehat{Y}}(y) p_S(s))^2}{p_{\widehat{Y}}(y) p_S(s)} \right) dy} \tag{29}$$

$$= \sqrt{D_R(\widehat{Y}; S)}, \tag{30}$$

where Eq. (28) follows from Cauchy-Schwarz inequality, and Eq. (30) follows from Lemma 2. The extension to general $Z$ and $\mathcal{Z}$ is immediate by observing that $\rho(\widehat{Y}, S | Z \in \mathcal{Z}) = \mathbb{E}_Z \left[ \rho(\widehat{Y}, S | Z) \middle| Z \in \mathcal{Z} \right]$, $\rho_R(\widehat{Y}, S | Z \in \mathcal{Z}) = \mathbb{E}_Z \left[ \rho_R(\widehat{Y}, S | Z) \middle| Z \in \mathcal{Z} \right]$, and $D_R(\widehat{Y}, S | Z \in \mathcal{Z}) = \mathbb{E}_Z \left[ D_R(\widehat{Y}, S | Z) \middle| Z \in \mathcal{Z} \right]$. □

**Lemma 2.** *We have*

$$D_R(\widehat{Y}; S) = \sum_{s \in \mathcal{S}} \int_{y \in \mathcal{Y}} \left( \frac{(p_{\widehat{Y},S}(y,s) - p_{\widehat{Y}}(y) p_S(s))^2}{p_{\widehat{Y}}(y) p_S(s)} \right) dy. \tag{31}$$

*Proof.* The proof follows from the following set of identities:

$$\sum_{s\in\mathcal{S}}\int_{y\in\mathcal{Y}}\left(\frac{(p_{\widehat{Y},S}(y,s)-p_{\widehat{Y}}(y)p_S(s))^2}{p_{\widehat{Y}}(y)p_S(s)}\right)dy = \sum_{s\in\mathcal{S}}\int_{y\in\mathcal{Y}}\frac{(p_{\widehat{Y},S}(y,s))^2}{p_{\widehat{Y}}(y)p_S(s)}dy$$

$$-2\sum_{s\in\mathcal{S}}\int_{y\in\mathcal{Y}}p_{\widehat{Y},S}(y,s)dy$$

$$+\sum_{s\in\mathcal{S}}\int_{y\in\mathcal{Y}}p_{\widehat{Y}}(y)p_S(s)dy \tag{32}$$

$$= E\left\{\frac{p_{\widehat{Y},S}(\widehat{Y},S)}{p_{\widehat{Y}}(\widehat{Y})p_S(S)}\right\} - 1 \tag{33}$$

$$= D_R(\widehat{Y};S). \tag{34}$$

$\square$

Next, we present some alternative fairness definitions and show that they are also upper bounded by ERMI.

**Definition 10** (conditional demographic parity $L_\infty$ violation). *Given a predictor $\widehat{Y}$ supported on $\mathcal{Y}$ and a discrete sensitive attribute $S$ supported on a finite set $\mathcal{S}$, we define the conditional demographic parity violation by:*

$$\widetilde{dp}(\widehat{Y}|S) := \sup_{\widehat{y}\in\mathcal{Y}}\max_{s\in\mathcal{S}}\left|p_{\widehat{Y}|S}(\widehat{y}|s)-p_{\widehat{Y}}(\widehat{y})\right|. \tag{35}$$

First, we show that $\widetilde{dp}(\widehat{Y}|S)$ is a reasonable notion of fairness violation.

**Lemma 3.** $\widetilde{dp}(\widehat{Y}|S) = 0$ *iff (if and only if) $\widehat{Y}$ and $S$ are independent.*

*Proof.* By definition, $\widetilde{dp}(\widehat{Y}|S) = 0$ iff for all $\widehat{y}\in\mathcal{Y}, s\in\mathcal{S}$, $p_{\widehat{Y},S}(\widehat{y}|s) = p_{\widehat{Y}}(\widehat{y})$ iff $\widehat{Y}$ and $S$ are independent (since we always assume $p(s) > 0$ for all $s\in\mathcal{S}$). $\square$

**Theorem 9** (ERMI is stronger than conditional demographic parity $L_\infty$ violation). *Let $\widehat{Y}$ be a discrete or continuous random variable supported on $\mathcal{Y}$, and $S$ be a discrete random variable supported on a finite set $\mathcal{S}$. Denote $p_S^{\min} := \min_{s\in\mathcal{S}} p_S(s) > 0$. Then,*

$$0 \leq \widetilde{dp}(\widehat{Y}|S) \leq \frac{1}{p_S^{\min}}\sqrt{D_R(\widehat{Y};S)}. \tag{36}$$

*Proof.* The proof follows from the following set of (in)equalities:

$$\left(\widetilde{dp}(\widehat{Y}|S)\right)^2 = \sup_{\widehat{y}\in\mathcal{Y}}\max_{s\in\mathcal{S}}\left(p_{\widehat{Y}|S}(\widehat{y}|s)-p_{\widehat{Y}}(\widehat{y})\right)^2 \tag{37}$$

$$\leq \frac{1}{(p_S^{\min})^2}\sup_{\widehat{y}\in\mathcal{Y}}\max_{s\in\mathcal{S}}\left(p_{\widehat{Y},S}(\widehat{y},s)-p_{\widehat{Y}}(\widehat{y})p_S(s)\right)^2 \tag{38}$$

$$\leq \frac{1}{(p_S^{\min})^2}\int_{\widehat{y}\in\mathcal{Y}}\sum_{s\in\mathcal{S}}\left(p_{\widehat{Y},S}(\widehat{y},s)-p_{\widehat{Y}}(\widehat{y})p_S(s)\right)^2 \tag{39}$$

$$= \frac{1}{(p_S^{\min})^2}D_R(\widehat{Y};S), \tag{40}$$

where Eq. (40) follows from Theorem 3. $\square$

**Definition 11** (conditional equal opportunity $L_\infty$ violation (Hardt et al., 2016))**.** *Let $Y, \widehat{Y}$ take values in $\mathcal{Y}$ and let $\mathcal{A} \subseteq \mathcal{Y}$ be a compact subset denoting the advantaged outcomes (For example, the decision "to interview" an individual or classify an individual as a "low risk" for financial purposes). We define the conditional equal opportunity $L_\infty$ violation of $\widehat{Y}$ with respect to the sensitive attribute $S$ and the advantaged outcome $\mathcal{A}$ by*

$$\widetilde{eo}(\widehat{Y}|S, Y \in \mathcal{A}) := \mathbb{E}_Y \left\{ \sup_{\widehat{y} \in \mathcal{Y}} \max_{s \in \mathcal{S}} \left| p_{\widehat{Y}, S|Y}(\widehat{y}|s, Y) - p_{\widehat{Y}|Y}(\widehat{y}|Y) \right| \middle| Y \in \mathcal{A} \right\}. \quad (41)$$

**Theorem 10** (ERMI is stronger than generalized equal opportunity Kolmogorov violation - alternative definition)**.** *Let $\widehat{Y}$, $Y$, be discrete or continuous random variables supported on $\mathcal{Y}$, and let $S$ be a discrete random variable supported on a finite set $\mathcal{S}$. Let $\mathcal{A} \subseteq \mathcal{Y}$ be a compact subset of $\mathcal{Y}$. Denote $p_{S|\mathcal{A}}^{\min} = \min_{s \in \mathcal{S}, y \in \mathcal{A}} p_{S|Y}(s|y)$. Then,*

$$0 \le \widetilde{eo}(\widehat{Y}|S, Y \in \mathcal{A}) \le \frac{1}{p_{S|\mathcal{A}}^{\min}} \sqrt{D_R(\widehat{Y}; S|Y \in \mathcal{A})}. \quad (42)$$

*Proof.* Notice that the same proof for Theorem 9 would give that for all $y \in \mathcal{A}$:

$$0 \le \sup_{\widehat{y} \in \mathcal{Y}} \max_{s \in \mathcal{S}} \left| p_{\widehat{Y}, S|Y}(\widehat{y}|s, y) - p_{\widehat{Y}|Y}(\widehat{y}|y) \right| := \widetilde{eo}(\widehat{Y}|S, Y = y)$$

$$\le \frac{1}{p_{S|y}^{\min}(y)} \sqrt{D_R(\widehat{Y}; S|Y = y)}$$

$$\le \frac{1}{p_{S|\mathcal{C}}^{\min}} \sqrt{D_R(\widehat{Y}; S|Y = y)}.$$

Hence,

$$\widetilde{eo}(\widehat{Y}|S, Y \in \mathcal{A}) = \mathbb{E}_Y \left\{ \widetilde{eo}(\widehat{Y}|S, Y) \middle| Y \in \mathcal{A} \right\}$$

$$\le \frac{1}{p_{S|\mathcal{A}}^{\min}} \mathbb{E}_Y \left\{ \sqrt{D_R(\widehat{Y}; S|Y)} \middle| Y \in \mathcal{A} \right\}$$

$$\le \frac{1}{p_{S|\mathcal{A}}^{\min}} \sqrt{\mathbb{E}_Y \left\{ D_R(\widehat{Y}; S|Y) \middle| Y \in \mathcal{A} \right\}}$$

$$= \frac{1}{p_{S|\mathcal{A}}^{\min}} \sqrt{D_R(\widehat{Y}; S|Y \in \mathcal{A})},$$

where the last inequality follows from Jensen's inequality. This completes the proof. $\qquad \square$

## D   STOCHASTIC FERMI

*Proof of Theorem 5.* Let $W^* \in \arg\max_{W \in \mathbb{R}^{k \times m}} -\operatorname{Tr}(W P_{\widehat{y}} W^T) + 2 \operatorname{Tr}(W P_{\widehat{y}, s} P_s^{-1/2})$. We will compute $W^*$ and plug it in the RHS of Eq. (16) to show the equality in Eq. (16). Setting the derivative of the expression on the RHS equal to zero leads to:

$$-2 W P_{\widehat{y}} + 2 P_s^{-1/2} P_{\widehat{y}, s}^T = 0 \implies W^* = P_{\widehat{y}}^{-1} P_{\widehat{y}, s}^T P_s^{-1/2}.$$

Plugging this expression for $W^*$, we have

$$\max_{W \in \mathbb{R}^{k \times m}} -\operatorname{Tr}(W P_{\widehat{y}} W^T) + 2 \operatorname{Tr}(W P_{\widehat{y}, s} P_s^{-1/2})$$

$$= -\operatorname{Tr}(P_s^{-1/2} P_{\widehat{y}, s}^T P_{\widehat{y}}^{-1} P_{\widehat{y}} P_{\widehat{y}}^{-1} P_s^{-1/2}) + 2 \operatorname{Tr}(P_s^{-1/2} P_{\widehat{y}, s}^T P_{\widehat{y}}^{-1} P_{\widehat{y}} P_{\widehat{y}}^{-1} P_s^{-1/2})$$

$$= \operatorname{Tr}(P_s^{-1/2} P_{\widehat{y}, s}^T P_{\widehat{y}}^{-1} P_{\widehat{y}, s} P_s^{-1/2})$$

$$= \operatorname{Tr}(P_s^{-1} P_{\widehat{y}, s}^T P_{\widehat{y}}^{-1} P_{\widehat{y}, s}).$$

Writing out the matrix multiplication explicitly in the last expression, we have

$$P_s^{-1} P_{\widehat{y},s}^T P_{\widehat{y}}^{-1} P_{\widehat{y},s} = UV^T,$$

where $U_{i,j} = \widehat{p}_S(i)^{-1} \widehat{p}_{\widehat{Y},S}(j,i)$ and $V_{i,j} = \widehat{p}_{\widehat{Y}}(j)^{-1} \widehat{p}_{\widehat{Y},S}(j,i)$, for $i \in [k], j \in [m]$. Hence

$$\max_{W \in \mathbb{R}^{k \times m}} - \text{Tr}(WP_{\widehat{y}}W^T) + 2\,\text{Tr}(WP_{\widehat{y},s}P_s^{-1/2}) = \text{Tr}(UV^T)$$

$$= \sum_{i \in [k]} \sum_{j \in [m]} \frac{p_{\widehat{Y},S}(j,i)^2}{p_S(i)p_{\widehat{Y}}(j)}$$

$$= D_R(\widehat{Y}; S),$$

which completes the proof. $\qquad\square$

Next, we move to the statement and proof of the precise version of Theorem 6. We first recall some basic definitions:

**Definition 12.** *A function $f$ is $\beta$-smooth if for all $\mathbf{u}, \mathbf{u}'$, we have $\|\nabla f(\mathbf{u}) - \nabla f(\mathbf{u})\| \leq \beta \|\mathbf{u} - \mathbf{u}'\|$.*

**Definition 13.** *A point $\boldsymbol{\theta}$ is an $\epsilon$-stationary point of a differentiable function $\Phi$ if $\|\nabla \Phi(\boldsymbol{\theta})\| \leq \epsilon$.*

**Assumption 2.**
- *$\ell$ is twice differentiable, $L_\ell$-Lipscthiz, and $\beta_\ell$-smooth in $\boldsymbol{\theta}$.*

- $\|\nabla_\theta P_{\widehat{y}}\|_2 := \|\nabla_\theta \text{vec}(P_{\widehat{y}})\|_2 \leq L_y$ *and* $\max_{l \in [m]} \|\nabla_\theta ((P_{\widehat{y}})_{l,l})\|_2 \leq \widetilde{L}_y$

- $\max_{l \in [m]} \|\nabla_{\theta\theta}^2 (P_{\widehat{y}})_{l,l}\|_2 \leq \beta_y.$

- $\|\nabla_\theta P_{\widehat{y},s}^T\|_2 := \|\nabla_\theta \text{vec}(P_{\widehat{y},s}^T)\|_2 \leq L_{ys}$ *and* $\max_{l \in [m], j \in [k]} \|\nabla_\theta ((P_{\widehat{y},s})_{l,m})\|_2 \leq \widetilde{L}_{ys}$

- $\max_{l \in [m], j \in [k]} \|\nabla_{\theta\theta}^2 (P_{\widehat{y},s})_{l,j}\|_2 \leq \beta_{y,s}.$

**Theorem 11** (Precise version of Theorem 6). *Denote*

$$f(\boldsymbol{\theta}, W) = \frac{1}{N} \sum_{i \in [N]} \ell(\mathbf{x}_i, y_i; \boldsymbol{\theta}) + \lambda \left( - \text{Tr}(WP_{\widehat{y}}W^T) + 2\,\text{Tr}(WP_{\widehat{y},s}P_s^{-1/2}) - 1 \right).$$

*Set $\mathcal{W} := B_F(0, 2D) \subset \mathbb{R}^{k \times m}$ (Frobenius norm ball of radius $2D$), $D := \frac{\sqrt{mk}}{\widehat{p}_{\widehat{y}}^{\min} \sqrt{\widehat{p}_s^{\min}}}$. Denote $\Delta_\Phi := \Phi(\theta_0) - \min_\theta \Phi(\theta)$, where $\Phi(\theta) := \max_{W \in \mathcal{W}} f(\boldsymbol{\theta}, W)$. In Algorithm 1, choose the step-sizes as $\eta_\theta = \Theta(1/\kappa^2\beta)$ and $\eta_W = \Theta(1/\beta)$ and mini-batch size as $M = \Theta\left(\max\left\{1, \kappa\sigma^2\epsilon^{-2}\right\}\right)$. Then under Assumption 2, the iteration complexity of Algorithm 1 to return an $\epsilon$-stationary point of $f$ is bounded by*

$$\mathcal{O}\left(\frac{\kappa^2\beta\Delta_\Phi + \kappa\beta^2 D^2}{\epsilon^2}\right),$$

*which gives the total stochastic gradient complexity of*

$$\mathcal{O}\left(\frac{\kappa^2\beta\Delta_\Phi + \kappa\beta^2 D^2}{\epsilon^2} \max\left\{1, \kappa\sigma^2\epsilon^{-2}\right\}\right),$$

*where*

$$\beta = \beta_l + 8\lambda D^2 \beta_y + 4\lambda \frac{1}{\sqrt{\widehat{p}_s^{\min}}} \left(\sqrt{m}k^{3/2}D\beta_{ys}\right) + 2\lambda + 4\lambda \left(DL_y + \frac{L_{ys}}{\sqrt{\widehat{p}_s^{\min}}}\right),$$

$$\mu = 2\lambda\widehat{p}_{\widehat{y}}^{\min},$$

$$\kappa = \beta/\mu,$$

$$\sigma^2 = 2\left(L_\ell + 2\lambda\widetilde{L}_y D^2 + 4\lambda\frac{D}{\sqrt{\widehat{p}_s^{\min}}}\sqrt{mk}\widetilde{L}_{ys}\right)^2 + 2\left(2\lambda D + 2(\widehat{p}_s^{\min})^{-1/2}\sqrt{mk}\right)^2$$

The theorem follows from (Lin et al., 2020, Theorem 4.5) combined with the following technical lemmas. We assume Assumption 2 holds for the remainder of the proof of Theorem 11:

**Lemma 4.** *Let*

$$f(\boldsymbol{\theta}, W) = \frac{1}{N} \sum_{i \in [N]} \ell(\mathbf{x}_i, y_i; \boldsymbol{\theta}) + \lambda \left( - \operatorname{Tr}(W P_{\widehat{y}} W^T) + 2 \operatorname{Tr}(W P_{\widehat{y},s} P_s^{-1/2}) - 1 \right)$$

$$:= \frac{1}{N} \sum_{i \in [N]} g(\boldsymbol{\theta}, W, \mathbf{x}_i, y_i).$$

*Then*

1. *$f$ is $\beta$-smooth, where $\beta = \beta_l + 8\lambda D^2 \beta_y + 4\lambda \frac{1}{\sqrt{\widehat{p}_s^{\min}}} \left( \sqrt{m} k^{3/2} D \beta_{ys} \right) + 2\lambda +$*

   *$4\lambda \left( D L_y + \frac{L_{ys}}{\sqrt{\widehat{p}_s^{\min}}} \right)$.*

2. *$f(\boldsymbol{\theta}, \cdot)$ is $2\lambda \widehat{p}_{\widehat{y}}^{\min}$-strongly concave for all $\boldsymbol{\theta}$.*

3. *$\|W^*\|_F \leq D$, where $D$ is as defined in Theorem 11 and $W^*$ denotes any maximizer of $f(\boldsymbol{\theta}, W)$.*

*Proof.* By Assumption 2, $g$ is twice continuously differentiable. Hence for part 1, it suffices to upper bound the spectral norm of the second derivative of $g(\cdot, \cdot, \mathbf{z})$ by $\beta$ for all $\mathbf{z} = (\mathbf{x}, y)$, where we vectorize and then differentiate with respect to $w := \operatorname{vec} W$ and/or $\boldsymbol{\theta}$, so that the resulting first and second derivatives are always vectors or a matrices (not tensors). Notice that $g(\boldsymbol{\theta}, w, \mathbf{z}) = \ell(\mathbf{z}, \boldsymbol{\theta}) - \lambda w^T (P_{\widehat{y}} \otimes \mathbf{I}) w + 2\lambda (\operatorname{vec}(W))^T P_{\widehat{y},s} P_s^{-1/2} - \lambda$ and

$$\nabla^2 g(\boldsymbol{\theta}, w, \mathbf{z}) = \begin{pmatrix} \nabla^2_{\theta\theta} g(\boldsymbol{\theta}, w, \mathbf{z}) & \nabla^2_{\theta w} g(\boldsymbol{\theta}, w, \mathbf{z}) \\ \nabla^2_{w\theta} g(\boldsymbol{\theta}, w, \mathbf{z}) & \nabla^2_{ww} g(\boldsymbol{\theta}, w, \mathbf{z}) \end{pmatrix}.$$

Further, by the definition of operator norm, we have

$$\|\nabla^2 g(\boldsymbol{\theta}, w, \mathbf{z})\|_2 \leq \|\nabla^2_{\theta\theta} g(\boldsymbol{\theta}, w, \mathbf{z})\|_2 + 2\|\nabla^2_{\theta w} g(\boldsymbol{\theta}, w, \mathbf{z})\|_2 + \|\nabla^2_{ww} g(\boldsymbol{\theta}, w, \mathbf{z})\|_2.$$

Now we vectorize all matrices and then compute derivatives of $g$ with respect to $\theta$ and $\operatorname{vec}(W)$:

$$\nabla_\theta g(\boldsymbol{\theta}, w, \mathbf{z}) = \nabla_\theta \ell(\mathbf{z}, \boldsymbol{\theta}) - 2\lambda \nabla_\theta \operatorname{vec}(P_{\widehat{y}})^T \operatorname{vec}(W^T W) + 2\lambda \nabla_\theta \operatorname{vec}(P_{\widehat{y},s})^T \operatorname{vec}(W^T P_s^{-1/2})$$

$$\tag{43}$$

$$= \nabla_\theta \ell(\mathbf{z}, \boldsymbol{\theta}) - 2\lambda \left[ \sum_{l \in [m], i \in [k]} W_{i,l}^2 \nabla_\theta \left( (P_{\widehat{y}})_{l,l} \right) \right]$$

$$+ 2\lambda \left[ \sum_{j \in [m], i \in [k]} W_{i,j} (\nabla_\theta (P_{\widehat{y}s})_{j,i}) (P_s^{-1/2})_{i,i} \right]; \tag{44}$$

$$\nabla_w g(\boldsymbol{\theta}, w, \mathbf{z}) = -2\lambda W P_{\widehat{y}} + 2\lambda P_s^{-1/2} P_{\widehat{y},s}^T. \tag{45}$$

Differentiating again yields:

$$\nabla^2_{ww} g(\boldsymbol{\theta}, w, \mathbf{z}) = -2\lambda P_{\widehat{y}} \otimes \mathbf{I}_k;$$

$$\nabla^2_{w\theta} g(\boldsymbol{\theta}, w, \mathbf{z}) = \frac{\partial}{\partial \theta} \frac{\partial g(\boldsymbol{\theta}, w, \mathbf{z})}{\partial w} = -2\lambda (\mathbf{I}_m \otimes W) \nabla_\theta P_{\widehat{y}} + 2\lambda (\mathbf{I}_m \otimes P_s^{-1/2}) \nabla_\theta \operatorname{vec}(P_{\widehat{y},s}^T);$$

$$\nabla^2_{\theta\theta} g(\boldsymbol{\theta}, w, \mathbf{z}) = \nabla^2_\theta \ell(\mathbf{z}, \boldsymbol{\theta}) - 2\lambda \left[ \sum_{l \in [m], i \in [k]} W_{i,l}^2 \nabla^2_{\theta\theta} \left( (P_{\widehat{y}})_{l,l} \right) \right]$$

$$+ 2\lambda \left[ \sum_{j \in [m], i \in [k]} W_{i,j} (\nabla^2_{\theta\theta} (P_{\widehat{y}s})_{j,i}) (P_s^{-1/2})_{i,i} \right].$$

Then to establish part 1, use Assumption 2, Clairaut's theorem, the definitions of the matrices and fact that their entries are in $[0, 1]$, the relations $\|AB\|_2 \leq \|A\|_2\|B\|_2$ and $\|\operatorname{vec} W\|_1 \leq \sqrt{mk}\|\operatorname{vec} W\|_2 = \sqrt{mk}\|W\|_F$, and the fact that $\|A \otimes B\|_2 = \|A\|_2\|B\|_2$ to bound the spectral norm of each second derivative above.

The strong concavity statement follows by noticing $\nabla^2_{ww}g(\boldsymbol{\theta}, W) \preccurlyeq -\mu\mathbf{I}$ iff $P_{\widehat{y}} \succcurlyeq \frac{\mu}{2\lambda}\mathbf{I}$ iff $\min_{i\in[m]} p_{\widehat{y}}(i) \geq \frac{\mu}{2\lambda}$.

Part 3 follows from the expression for $W^*$ in the proof of Theorem 5. $\qquad\square$

**Lemma 5.** *Consider $f$ and $g$ as defined above. Then we have*

$$\mathbb{E}_{\mathbf{z}}\nabla g(\boldsymbol{\theta}, W, \mathbf{z}) = \nabla f(\boldsymbol{\theta}, W),$$

$$\mathbb{E}_{\mathbf{z}}\|\nabla g(\boldsymbol{\theta}, W, \mathbf{z}) - \nabla f(\boldsymbol{\theta}, W)\|_2^2 \leq 2\left(L_\ell + 2\lambda\widetilde{L}_y D^2 + 4\lambda\frac{D}{\sqrt{\hat{p}_s^{\min}}}\sqrt{mk}\widetilde{L}_{ys}\right)^2$$
$$+ 2\left(2\lambda D + 2(\hat{p}_s^{\min})^{-1/2}\sqrt{mk}\right)^2,$$

*where both expectations are with respect to the empirical distribution on $\{\mathbf{z}_i\}_{i\in[N]}$.*

*Proof.* The first statement is obvious. The second follows from Eq. (44) in the proof of Lemma 4, since

$$\mathbb{E}_{\mathbf{z}}\|\nabla g(\boldsymbol{\theta}, W, \mathbf{z}) - \nabla f(\boldsymbol{\theta}, W)\|_2^2$$
$$= \frac{1}{N}\sum_{i=1}^{N}\|\nabla g(\boldsymbol{\theta}, W, z_i)\|_2^2 - \frac{1}{N^2}\sum_{i,j=1}^{N}\langle\nabla g(\boldsymbol{\theta}, W, \mathbf{z}_i), \nabla g(\boldsymbol{\theta}, W, \mathbf{z}_j)\rangle$$
$$\leq 2\sup_{\mathbf{z}_i}\|\nabla g(\boldsymbol{\theta}, W, \mathbf{z}_i)\|_2^2$$
$$\leq 2\sup_{\mathbf{z}}\left\{\|\nabla_\theta g(\boldsymbol{\theta}, W, \mathbf{z})\|^2 + \|\nabla_w g(\boldsymbol{\theta}, W, \mathbf{z})\|^2\right\}$$
$$\leq 2\sup_{\mathbf{z}}\left\{\left\|\nabla_\theta\ell(\mathbf{z}, \boldsymbol{\theta}) - 2\lambda\left[\sum_{l\in[m],i\in[k]}W_{i,l}^2\nabla_\theta\left((P_{\widehat{y}})_{l,l}\right)\right]\right.\right.$$
$$\left.\left. + 2\lambda\left[\sum_{j\in[m],i\in[k]}W_{i,j}(\nabla_\theta(P_{\widehat{y}s})_{j,i})(P_s^{-1/2})_{i,i}\right]\right\|_2^2\right\}$$
$$+ 2\left\|-2\lambda WP_{\widehat{y}} + 2\lambda P_s^{-1/2}P_{\widehat{y},s}^T\right\|_2^2.$$

Then use Assumption 2 and basic norm inequalities to bound the norm of each term. $\qquad\square$

# E  EXPERIMENT DETAILS

## E.1  MODEL DESCRIPTION

For all the experiments, the model's output is of the form $O = \operatorname{softmax}(Wx + b)$. The model outputs are treated as conditional probabilities $\mathbf{p}(\widehat{y} = i|x) = O_i$ which are then used to estimate the ERMI regularizer. We encode the true class label $Y$ and sensitive attribute $S$ using one-hot encoding. We define $\ell()$ as the cross-entropy measure between the one-hot encoded class label $Y$ and the predicted output vector $O$.

We perform the experiments in sections 5.1 and 5.2 with a linear model (with softmax activation). The model parameters are estimated using the algorithm described in section 4.1. In section 5.2, the data set is cleaned and processed as described in (Kearns et al., 2018). The trade-off curves for FERMI are generated by sweeping across different values for $\lambda$ in [0, 100], learning rate $\eta$ in [0.0005, 0.01], and number of iterations $T$ in [50, 200].

For the experiments in section 5.3, we create the synthetic color MNIST as described in (Li & Vasconcelos, 2019). We set the value $\sigma = 0$. In figure 4, we compare the performance of stochastic solver (section 4.2) against the GD algorithm (section 4.1). We use a mini-batch of size 512 when using the stochastic solver. The color MNIST data has 60000 training samples, so using the stochastic solver gives a speedup of around 100x for each iteration, and an overall speedup of around 40x. We present our results on two neural network architectures; namely, LeNet-5 (Lecun et al., 1998) and a Multi-layer perceptron (MLP). We set the MLP with two hidden layers (with 300 and 100 nodes) and an output layer with ten nodes. A ReLU activation follows each hidden layer, and a softmax activation follows the output layer.

Some general advice for tuning $\lambda$: Larger value for $\lambda$ generally translates to better fairness, but one must be careful to not use a very large value for $\lambda$ as it could lead to poor generalization performance of the model. The optimal values for $\lambda$, $\eta$, and $T$ largely depend on the data and intended application. We recommend starting with $\lambda \approx 10$. In Appendix E.2, we can observe the effect of changing $\lambda$ on the model accuracy and fairness for the COMPAS dataset.

### E.2 EFFECT OF HYPER-PARAMETER TUNING ON THE ACCURACY-FAIRNESS TRADE-OFF

We run ERMI algorithm for the binary case to COMPAS dataset to investigate the effect of hyper-parameter tuning on the accuracy-fairness trade-off of the algorithm. As it can be observed in Fig. 5, by increasing $\lambda$ from 0 to 1000, test error (left axis, red curves) is slightly increased. On the other hand, the fairness violation (right axis, green curves) is decreased as we increase $\lambda$ to 1000. Moreover, for both notions of fairness (demographic parity with the solid curves and equality of opportunity with the dashed curves) the trade-off between test error and fairness follows the similar pattern. To measure the fairness violation, we use demographic parity violation and equality of opportunity violation defined in Section equation 5 for the solid and dashed curves respectively.

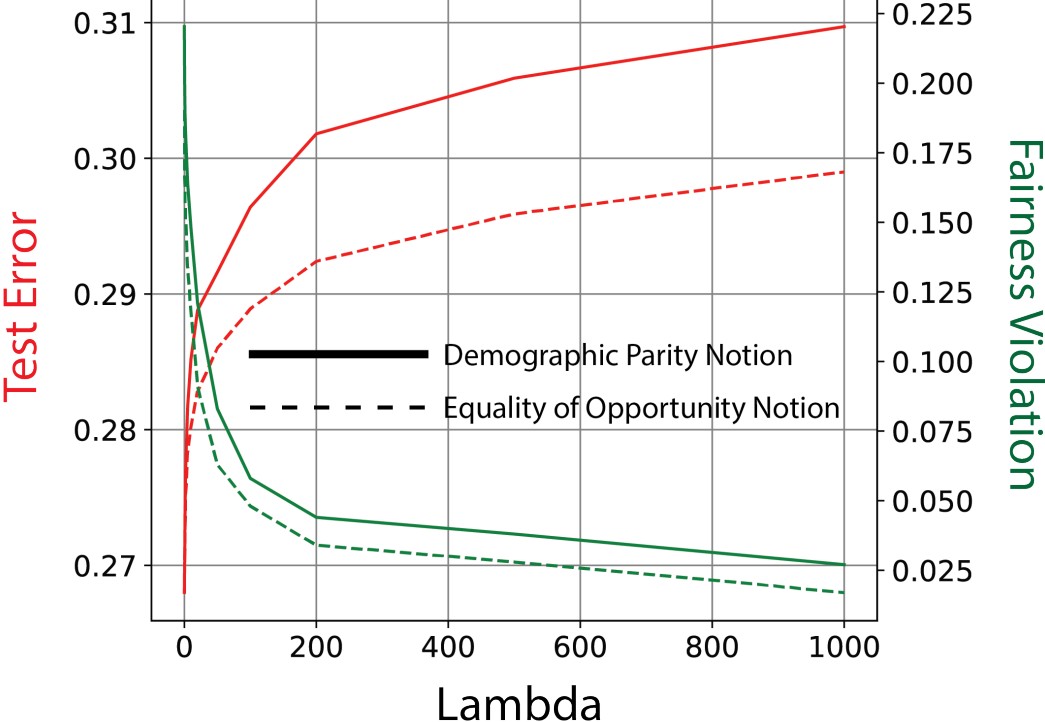

Figure 5: Tradeoff of fairness violation vs test error for ERMI algorithm on COMPAS dataset. The solid and dashed curves correspond to ERMI algorithm under the demographic parity and equality of opportunity notions accordingly. The left axis demonstrates the effect of changing $\lambda$ on the test error (red curves), while the right axis shows how the fairness of the model (measured by equality of opportunity or demographic parity violations) depends on changing $\lambda$.

### E.3  DATASETS DESCRIPTION

All of the following datasets are publicly available at UCI repository.

**German Credit Dataset.**[3]  German Credit dataset consists of 20 features (13 categorical and 7 numerical) regarding to social, and economic status of 1000 customers. The assigned task is to classify customers as good or bad credit risks. Without imposing fairness, the DP violation of the trained model is larger than 20%. We chose first 800 customers as the training data, and last 200 customers as the test data. The sensitive attributes are gender, and marital-status.

**Adult Dataset.**[4]  Adult dataset contains the census information of individuals including education, gender, and capital gain. The assigned classification task is to predict whether a person earns over 50k annually. The train and test sets are two separated files consisting of 32,000 and 16,000 samples respectively. We consider gender and race as the sensitive attributes (For the experiments involving one sensitive attribute, we have chosen gender). Learning a logistic regression model on the training dataset (without imposing fairness) shows that only 3 features out of 14 have larger weights than the gender attribute. Note that removing the sensitive attribute (gender), and retraining the model does not eliminate the bias of the classifier. the optimal logistic regression classifier in this case is still highly biased. For the clustering task, we have chosen 5 continuous features (Capital-gain, age, fnlwgt, capital-loss, hours-per-week), and 10,000 samples to cluster. The sensitive attribute of each individual is gender.

**Communities and Crime Dataset**.[5]  The dataset is cleaned and processed as described in (Kearns et al., 2018). Briefly, each record in this dataset summarizes aggregate socioeconomic information about both the citizens and police force in a particular U.S. community, and the problem is to predict whether the community has a high rate of violent crime.

**COMPAS Dataset**.[6]  Correctional Offender Management Profiling for Alternative Sanctions (COMPAS) is a famous algorithm which is widely used by judges for the estimation of likelihood of re-offending crimes. It is observed that the algorithm is highly biased against the black defendants. The dataset contains features used by COMPAS algorithm alongside with the assigned score by the algorithm within two years of the decision.

---

[3]`https://archive.ics.uci.edu/ml/datasets/statlog+(german+credit+data)`
[4]`https://archive.ics.uci.edu/ml/datasets/adult.`
[5]`http://archive.ics.uci.edu/ml/datasets/communities+and+crime`
[6]`https://www.kaggle.com/danofer/compass`

