# OpenReview forum: "FERMI: Fair Empirical Risk Minimization via Exponential Rényi Mutual Information"
_ICLR.cc/2021/Conference — Reject_

### Official Review · AnonReviewer2 · 2020-10-28
**Good work, provides a unified view of fairness violation notions and propose a measure to upperbound quite a few, good algorithmic contribution and relatively solid experiments.**

**Rating:** 7
**Confidence:** 4

**Review:**

The authors provide a unified view of the existing fairness violation notions and propose a new notion: Exponential Rényi Mutual Information (ERMI) between sensitive attributes and the label. ERMI is easy to compute and provides an upper bound on existing notions. Based on ERMI, the authors propose a framework, FERMI, which can be optimized with SGD with convergence guarantee. In experiments, results show that FERMI leads to a better tradeoff between performance and fairness even if fairness is not measured by ERMI.

My major concern about the methodology is that it depends on the quality of the empirical estimate of the probability distributions P(s), P(y) and P(y,s). In the experiments, these distributions are relatively simple as S takes either binary values or are samples from Gaussian distribution. And Y is discrete and takes few values. I wonder how the performance of the FERMI would be influenced by the quality of estimates of the three probability distributions. It would be interesting if the authors can show results on the estimation error of the three vs. the performance of FERMI.

In section 3, the authors show that ERMI is an upperbound of several popular notions: Shannon MI, Renyi correlation, and Lq fairness violation. I have several suggestions and questions: (1) It would be better to specify whether some of them are known before this work. (2) In addition, is there any other transformation of Renyi Mutual Information that can achieve the same goal? (3) Is it possible to specify how tight those bounds are?

In 5.1, the authors mentioned that one reason FERMI outperforms baselines is that ERMI upperbounds the other notions. I would appreciate it if the authors can clarify this point.

---

> ### Author Response · Authors · 2020-11-19
> **Thanks for your thoughtful review!**
>
> **[empirical estimates of P(s), P(y) and P(y,s)]** The reviewer is correct in observing that the way the batch algorithm is set up, the required sample complexity scales linearly with the cardinality of $\cal S$ and $\cal Y$. Having said that, we show in Figure 2 and experiment with $\cal S = \{2^{18}\}$. On the other hand, we also note that the stochastic algorithm does not require consistency of such estimates and an unbiased estimate suffices for convergence guarantees. We added a discussion in Section 4.1 to note the linear dependence on $|{\cal S}|$ and $|{\cal Y}|$ (i.e., exponential dependence on the number of sensitive attributes) as well as connecting it with the stochastic algorithm.
>
> **[point out known results]** To the best of our knowledge Theorems 1/2/3 are not observed in the literature. We have made connections with existing pieces and lemmas that we were aware of. Having said that, since ERMI is inherently an $f$-divergence, we would not be surprised if there are other pieces of related work that we may have missed.
>
> **[ In addition, is there any other transformation of Renyi Mutual Information that can achieve the same goal?]**
> We believe that regularization with Renyi mutual information of order 2 should potentially give the same fairness violation/performance tradeoff as ERMI given that ERMI and Renyi mutual information of order 2 are related via a smooth monotone mapping. However, one of the big advantages of ERMI is that per Theorem 5, it is amenable to stochastic optimization leading to our stochastic version of the FERMI algorithm (Algorithm 1) that could be used within the modern stochastic optimization framework of large-scale neural architectures.
>
> **[tightness of bounds]** We answered this question in our response to Reviewer 3. We repeat our response here for completeness: Conceptually the difference between these various notions of fairness is similar to the difference between various types of norms (e.g., $L_1$, $L_2$, $L_\infty$). While different norms will bound each other, these bounds could be arbitrarily loose for large alphabet sizes, which we suspect would be the case for our results as well. The slack in the bounds is natural in our context and indeed not the point. Notice that there is no universally accepted notion of fairness violation in the literature. The existing notions of fairness violation are not necessarily comparable (e.g., mutual information and conditional demographic parity $L_\infty$ violation). This raises the question of which one (if any) to select in practice for imposing fairness.  A natural approach is to use a violation notion that subsumes (bounds from above) all of these existing notions of fairness violation. Using a regularizer that bounds all notions leads to a trained model that is almost universally accepted under different notions of fairness proposed in the community. In addition to this benefit, from an optimization viewpoint, FERMI regularizer is smooth and is amenable to stochastic optimization (as shown in Section 4). These properties will allow the scalable training of ML models using FERMI regularizer. This is in contrast to many other regularizers in the literature, such as RFI in (Baharlouei et al. 2020), that are non-smooth (and also not amenable to stochastic implementation) which prevents them to be applied in large-scale problems.
>
> **[performance of FERMI]** We added a discussion to Section 6 to clarify this point.

---

### Official Review · AnonReviewer4 · 2020-10-28
**Confusing notations**

**Rating:** 5
**Confidence:** 3

**Review:**

The paper studies fair classification by using the notion of Exponential Renyi Mutual Information. As certain notions of fairness can be encoded using (conditional) independence, this paper propose to use some information theory notions of mutual information to quantify this degree of (conditional) independence, which indirectly translates to the degree of fairness. A classification algorithm with low value of mutual information between the prediction and the sensitive attributes can be considered as fair.

The paper establishes that the Exponential Renyi Mutual Information is a strong notion to ensure fairness: the authors show that this notion is stronger than many existing notions such as Lq fairness, etc. The authors propose an algorithm to train a fair classifier, with the mutual information being penalized in the objective function.

Strength:
- The idea of combining information theory notions to measure independency and applying it to fair machine learning is natural.

Weakness:
- I have some concerns with the mathematical notations in this paper.
i) The definition 2 is very misleading. Consider for equalized odds with \mathcal Z = \{0, 1\}: it is not clear to me why this definition correctly capture the conditional independence. To my best understanding, the conditioning here should be taken as 2 separate conditional expectation: one conditional expectation with Z = 0, and another conditional expectation with Z = 1. The mathematical definition in equation (2) does not seem to segregate the values of Z. Moreover, why is the expectation taken over Z when we already condition on Z \in \mathcal Z?
ii) What is D_R(\hat Y_\theta, S) in equation (11)? I guess the authors mean the D_R function with condition as in equation (2)?

- The authors show that ERMI is stronger than existing notions, which is nice. However, it is not clear why a stronger notion is preferable for the penalized optimization of the form (11). One can think of penalizing the Shannon mutual information with higher penalty parameter lambda, and one may expect to see similar end results as problem (11) -- especially if we plot the accuracy-fairness tradeoff similar to Figure 1.

Minor comments:
- Lemma 1 is quite trivial. I think this lemma should be put as discussion in the paper, and not a separate lemma.
- Why are the expectation in equation (15) taken with X?
- The proof of Theorem 5 does not seem to prove what is stated in Theorem 5.
- The notations in equation (11) can be made more explicit. For example, I think there is a dependence of \hat Y_\theta on X which is not made explicit.

---

> ### Author Response · Authors · 2020-11-19
> **Thanks for your thoughtful review!**
>
> **[concerns with notation]** Thanks for these suggestions. We now show how Definition 2 specializes to the cases of interest when the fairness notion of interest is demographic parity, equalized odds, or equal opportunity in Appendix B. To the specific point of the reviewer, please note that $I(X;Y|Z)$ is also defined in a similar manner to ERMI. In this case, $I(X;Y|Z) = 0$ implies the conditional independence of $X$ and $Y$ given $Z$. ERMI has a similar implication because ERMI is an $f$-divergence, and this general property holds for all $f$-divergences, as discussed in Appendix B.
>
> **[D_R(\hat Y_\theta, S)]** We thank the reviewer for this comment; we now explicitly define this notion in Eq. (17) and have added a note to the definition of FRMI for the connection.
>
> **[similar tradeoff with other notions of fairness]** We agree with the reviewer that other notions of fairness violation may potentially lead to a similar tradeoff curve. For example, given that ERMI and Renyi mutual information of order 2 are related via a bijective mapping, we suspect that their tradeoff curve should be similar under certain conditions. Having said that, a big advantage of ERMI is that we are able to develop a stochastic solver for it through Theorem 5, and apply it to large-scale optimization problems. This property clearly distinguishes ERMI from the rest of the notions.
>
> **[performance improvement not clear]** We added a discussion to Section 6 to address these points.
>
> **[Lemma 1 be moved to paper]** We followed the reviewers’ comment in the revised paper.
>
> **[expectations in (15)]** We corrected this equation to capture the right notion in the revised version.
>
> **[Proof of Theorem 5]** Thanks for pointing this out. We have updated the statement of Theorem 5 and its proof to clarify the connection.
>
> **[Explicit dependence of \hat Y_\theta on X]** We made that point explicit per reviewer’s suggestion.

---

### Official Review · AnonReviewer3 · 2020-10-31
**A review**

**Rating:** 5
**Confidence:** 4

**Review:**

Summary:

This paper introduces a new fairness notion that takes an exponential form of Renyi mutual information. The strongness of the notion is claimed via the proof (Theorems 1/2/3) that the notion is an upper bound of other well-known notions like mutual information, Renyi mutual information, Renyi correlation, L1 fairness violation. Two methods for estimating the notion via samples are developed under some assumptions. The performances of the methods are then demonstrated on three datasets and compared with some prior algorithms.

Strengths:

1. An interesting observation is made via Theorems 1/2/3: ERMI is an upper bound of other popular notions.
2. Explicit algorithms are developed for estimating the notion as differential functions w.r.t. model parameters.

Weaknesses:

1. While the observation in Theorems 1/2/3 is interesting, the proof relies upon the techniques in well-known literature and/or some standard tricks. Perhaps more importantly, it is not investigated in depth how tight the proposed notion is relative to other notions, except for the binary classification case. I believe the tightness analysis is more important, as a loose upper bound plays a less role.

2. Algorithm 1: No justification is provided behind the assumption, in particular (13). Also no analysis is provided regarding the accuracy of the density estimates (16). In particular, this reviewer wonders whether or not the condition density estimate (the 2nd in (16)) is accurate – usually, a precise accuracy requires an exponential number of samples w.r.t. the cardinality of yhat.

3. Experiments: Some baselines based on mutual information are missing, e.g., [A], [B]. Also in the 1st experiment (Fig. 1), (Baharlouei et al 2020) is missing – I think it is needed to compare, since the notion estimation method may be different although ERMI is equivalent to Renyi correlation. Moreover, it is not clear which algorithm to use in all experiments between Algorithms 1 and 2.

[A] B. Zhang, B. Lemoine, and M. Mitchell. Mitigating unwanted biases with adversarial learning. AIES, 2018.
[B] J. Cho, G. Hwang and C. Suh, “A fair classifier using mutual information,” ISIT 2020.

Clarity: Overall it is well-written and easy to follow.

Other comments:
i. Why italic in the last sentence in the 2nd paragraph on page 1?
ii. Details on hyperparameter tuning are entirely missing in all experiments.
iii.  A benchmark dataset, COMPAS, is missing.
iv. Not clear why the proposed algorithms offer greater performances. No insight/analysis is provided.
v. Figure 2: Only RFI is compared, although many other baselines can readily be extentisible to the non-binary classification.
iv. Figure 4: Not clear as to what points the authors wish to make here.

---

> ### Author Response · Authors · 2020-11-19
> **Thanks for your thoughtful review!**
>
> **[Novelty of Theorems 1/2/3]** We note that we are unaware of any existing literature making the connections in Theorems 1/2/3, and we believe that these results are novel. We point to the existing literature for the already known lemmas and techniques.
>
> **[Tightness of Theorems 1/2/3]** Conceptually the difference between these various notions of fairness is similar to the difference between various types of norms (e.g., $L_1$, $L_2$, $L_\infty$). While different norms will bound each other, these bounds could be arbitrarily loose for large alphabet sizes, which we suspect would be the case for our results as well. The slack in the bounds is natural in our context and indeed not the point. Notice that there is no universally accepted notion of fairness violation in the literature. The existing notions of fairness violation are not necessarily comparable (e.g., mutual information and conditional demographic parity $L_\infty$ violation). This raises the question of which one (if any) to select in practice for imposing fairness.  A natural approach is to use a violation notion that subsumes (bounds from above) all of these existing notions of fairness violation. Using a regularizer that bounds all notions leads to a trained model that is almost universally accepted under different notions of fairness proposed in the community. In addition to this benefit, from an optimization viewpoint, FERMI regularizer is smooth and is amenable to stochastic optimization (as shown in Section 4). These properties will allow the scalable training of ML models using FERMI regularizer. This is in contrast to many other regularizers in the literature, such as RFI in (Baharlouei et al. 2020), that are non-smooth (and also not amenable to stochastic implementation) which prevents them to be applied in large-scale problems.
>
> **[Justification of (13)]** Equation (13) is natural in problem instances where the decisions made by the learning algorithm are soft decisions. This covers a wide range of classification methods such as logistic regression or classification via neural networks with the last layer being the popular softmax layer. Having said that, our general framework is not limited to this choice and as long as we can take the derivative of the objective in (12) w.r.t. \theta, one can still rely on our framework. We included this discussion in the paper.
>
> **[Algorithm 1]** Your concern about the empirical estimates in (16) (equation (14) in the new version) is valid for the batch algorithm, and we added a discussion around this in Section 4.1. However, Algorithm 1 is a stochastic algorithm and is applicable with a batch size as small as 1. In this case, we only need an unbiased estimate of those said densities for the algorithm to converge, which is provided by the empirical estimates. We added this explanation after equation (16) in the paper.
>
> **[Experiments]** In the binary classification case with a binary sensitive attribute, not only ERMI is equal to Renyi correlation, but also our batch algorithm is exactly equivalent to that of RFI (Baharlouei et al. 2020). While we had already stated this in Section 4.1, we also added a note in Section 5.1 to make this clear. Thanks for providing the other two mutual information-based references, which we have included in citations. We also included [A] in the experimental results but were unable to find an implementation of [B] to include in the results of this section.
> [A] B. Zhang, B. Lemoine, and M. Mitchell. Mitigating unwanted biases with adversarial learning. AIES, 2018.
> [B] J. Cho, G. Hwang and C. Suh, “A fair classifier using mutual information,” ISIT 2020.
>
> **[Why italic in the last sentence in the 2nd paragraph on page 1?]** Given the high stakes of improper use of notions of algorithmic fairness in practice, we believe it is important to highlight the shortcomings of such methods upfront for practitioners.
>
> **[Details of hyperparameter tuning]** We added these in Section E.1.
>
> **[COMPAS benchmark]** While we report experiments on three other widely used benchmarks, we are happy to include COMPAS as well if the reviewer thinks that would add value to the paper for the reader.
>
> **[Performance of FERMI]** We added a discussion in Section 6.
>
> **[Figure 2: Only RFI is compared]** While some existing notions of fairness can be generalized to non-binary sensitive attributes or non-binary classification, we are unaware of baselines that readily generalize to both non-binary sensitive attributes and non-binary classification. We would appreciate it if the reviewer can point us to the relevant works and we are happy to include them in Figure 2.
>
> **[Figure 4: the point is unclear]** The main point of this experiment is to showcase the scaling of the stochastic FERMI, and the applicability of FERMI when the models of interest are large-scale neural networks.

---

### Author Response · Authors · 2020-11-19
**General response to all reviewers**

We would like to thank all reviewers for their valuable comments that provided us with the opportunity to improve the clarity of the presentation and mathematical exposition of the paper. Here, we address shared concerns and then respond to each of the reviewer’s comments individually

**[contribution]** We propose ERMI as a strong notion of fairness violation that provides guarantees on other existing notions of fairness violation (Theorems 1-3). In addition, ERMI is smooth, and more importantly, it is amenable to stochastic optimization (Theorem 5). This led us to the development of batch and stochastic methods for solving empirical risk minimization with an ERMI regularizer (FERMI). The stochastic solver is particularly important because it is applicable to large-scale optimization problems with neural network models (and we provide one such experiment in Section 5.3). We are unaware of any other in-process algorithm that could be applied to such large-scale problems. From a practical perspective, we observe that FERMI leads to a better tradeoff between performance and fairness violation even when such violation is measured in notions other than ERMI. We end the paper with a discussion on the potential reasons for this superior performance.

**[tightness of the bounds]** Several reviewers ask about the tightness of the bounds in Theorems 1-3. We believe that it should be possible to construct examples where these bounds are arbitrarily loose for large alphabet sizes, similarly to how $L_q$ norms are related with each other for different values of $q$. However, that is beside the point of these bounds. Currently, there is no agreed upon notion of fairness violation in the community. These guarantees motivate the use of ERMI as a unifying notion of fairness, because Theorems 1-3 shows that by providing guarantees on ERMI, one can obtain guarantees on all other notions of fairness violation.

**[sample complexity requirements]** Several reviewers question the sample complexity requirements for naively estimating ERMI from samples in the batch algorithm. While we agree with the reviewers that the sample complexity of estimating the distributions $p(s)$, $p(\hat{y})$, and $p(s, \hat{y})$ scales exponentially with the number of the sensitive attributes, we have developed stochastic FERMI for large-scale problems. We notice that stochastic FERMI (Algorithm 1) developed in Section 4.3 does not require such sample complexity as it only requires an unbiased estimator of the distributions of sensitive attributes and targets for it to converge. It remains an open question on how many samples it requires to converge to the exponential Renyi mutual information but we suspect such sample complexity is more favorable than other notions of fairness violation measures due to the existing results on the estimation of Renyi entropy of order 2 [1].

[1] Acharya, J., Orlitsky, A., Suresh, A.T. and Tyagi, H., 2014, December. The complexity of estimating Rényi entropy. In Proceedings of the twenty-sixth annual ACM-SIAM symposium on Discrete algorithms (pp. 1855-1869). Society for Industrial and Applied Mathematics.

**[superiority of FERMI]**
One possible reason is that the objective function is easier to optimize than the objectives of competing in-processing methods: ERMI is smooth; and in the discrete case, is equal to the trace of a matrix (see Theorem 8), which is easy to compute. Contrast this with the larger computational overhead of Renyi correlation, for example, which requires finding the second singular value of a matrix. Furthermore, the sample complexity of estimating Renyi mutual information of order $2$ (and consequently that of ERMI) scales as $\Theta(\sqrt{|{\cal S}|})$ as compared to Shannon mutual information which scales as $\Theta(|{\cal S}|/\log |{\cal S}|)$ [1]. Another possible explanation is that ERMI is a stronger notion of fairness than all of the most widely used fairness notions, as shown in Sec. 3, which might lead to better generalization. Together, these facts suggest that ERMI serves as an efficient and easily optimizable proxy for these other notions, leading to better practical performance regardless of which fairness measure is used. We leave it as future work to rigorously understand  which of these (or other) factors are responsible for the favorable performance tradeoffs observed from FERMI. Finally, on the Color MNIST experiment with neural network function approximation, we observed that stochastic FERMI outperforms batch FERMI. In this case, we suspect that the randomness in stochastic FERMI supposedly contributes to its convergence to a local minimum with superior generalization performance compared to batch FERMI (see [2] and the references therein).

[2] Kleinberg, B., Li, Y. and Yuan, Y., 2018, July. An Alternative View: When Does SGD Escape Local Minima?. In International Conference on Machine Learning (pp. 2698-2707).

---

### Decision · Program_Chairs · 2021-01-07
**Final Decision**

**Decision:**

Reject

**Comment:**

The paper proposed a new in-processing approach to train fair predictors under several notions of statistical fairness. Tho this end, the author rely  on  the Exponential Renyi Mutual Information (ERMI) between sensitive attributes and the target variable as notion f fairness, and show that it is a strong notion of fairness that provides guarantees on several previously discussed fairness metrics.

The paper is overall well written and interesting, but as with many other papers on this area, I wonder even after rebuttal whether the paper indeed constitute a step forward in the field. I find the concern raised by the reviewers about the tightness of the bound important and, while the authors properly addressed this point in the rebuttal period, I still believe this is an open question which probably does not have a better answer. On the positive side, the experimental evaluation support the theoretical results. However, comparisons to previous methods are only performed on the Adult and the German dataset, which makes me wonder if the advantages of the proposed approach generalize beyond these two well-studied datasets. As a consequence, the paper remains borderline, as it is an interesting paper but its impact and significance remain limited.

Moreover, I believe that there are some missing recent related works, that I believe the authors should also compare to. For example, see the recent Neurips 2020 paper, "A Fair Classifier Using Kernel Density Estimation" by Cho et al.  Also, as a side note, previous approached have already considered non-binary (although most of the times categorical) sensitive features, see e.g., [42]. Finally, the author may want to consider complementing their italic comment on the second paragraph of the Intro with existing works that already discussed biased in the labels, due to e.g., the selective labeling problem (see [1-3] below).

[1] Lakkaraju, Himabindu, et al. "The selective labels problem: Evaluating algorithmic predictions in the presence of unobservables." Proceedings of the 23rd ACM SIGKDD International Conference on Knowledge Discovery and Data Mining. 2017.
[2] Kilbertus, Niki, et al. "Fair decisions despite imperfect predictions." International Conference on Artificial Intelligence and Statistics. PMLR, 2020.
[3] Bechavod, Yahav, et al. "Equal opportunity in online classification with partial feedback." Advances in Neural Information Processing Systems. 2019.

---

> ### Author Response · Authors · 2021-01-26
> **Reference on non-binary classification with non-binary sensitive attributes**
>
> Dear AC,
>
> Thanks for your thorough review.
>
> The previous work by Zafar et al. mentioned in your comment considers non-binary sensitive features for a **binary** classification problem. We wonder if you are aware of any previous work that can simultaneously handle both non-binary sensitive features and non-binary classification.
>
> Authors.